# Chr21 protein–protein interactions: enrichment in proteins involved in intellectual disability, autism, and late-onset Alzheimer's disease

Julia Viard[1,2,*], Yann Loe-Mie[1,*], Rachel Daudin[1], Malik Khelfaoui[1], Christine Plancon[2], Anne Boland[2], Francisco Tejedor[3], Richard L Huganir[4], Eunjoon Kim[5], Makoto Kinoshita[6], Guofa Liu[7], Volker Haucke[8], Thomas Moncion[9], Eugene Yu[10], Valérie Hindie[9], Henri Bléhaut[11], Clotilde Mircher[11], Yann Herault[12,13,14,15,16], Jean-François Deleuze[2], Jean-Christophe Rain[9], Michel Simonneau[1,17,18,**], Aude-Marie Lepagnol-Bestel[1,**]

Down syndrome (DS) is caused by human chromosome 21 (HSA21) trisomy. It is characterized by a poorly understood intellectual disability (ID). We studied two mouse models of DS, one with an extra copy of the *Dyrk1A* gene (189N3) and the other with an extra copy of the mouse Chr16 syntenic region (Dp(16)1Yey). RNA-seq analysis of the transcripts deregulated in the embryonic hippocampus revealed an enrichment in genes associated with chromatin for the 189N3 model, and synapses for the Dp(16)1Yey model. A large-scale yeast two-hybrid screen (82 different screens, including 72 HSA21 baits and 10 rebounds) of a human brain library containing at least $10^7$ independent fragments identified 1,949 novel protein–protein interactions. The direct interactors of HSA21 baits and rebounds were significantly enriched in ID-related genes (*P*-value $< 2.29 \times 10^{-8}$). Proximity ligation assays showed that some of the proteins encoded by HSA21 were located at the dendritic spine postsynaptic density, in a protein network at the dendritic spine postsynapse. We located HSA21 DYRK1A and DSCAM, mutations of which increase the risk of autism spectrum disorder (ASD) 20-fold, in this postsynaptic network. We found that an intracellular domain of DSCAM bound either DLGs, which are multimeric scaffolds comprising receptors, ion channels and associated signaling proteins, or DYRK1A. The DYRK1A-DSCAM interaction domain is conserved in *Drosophila* and humans. The postsynaptic network was found to be enriched in proteins associated with ARC-related synaptic plasticity, ASD, and late-onset Alzheimer's disease. These results highlight links between DS and brain diseases with a complex genetic basis.

## Introduction

Down syndrome (DS) is the most common form of intellectual disability (ID). Its prevalence is influenced by maternal age at conception, which varies between countries, and has been estimated at ~1 in 365 fetuses at 10 wk of gestation (Antonarakis et al, 2020). This human genetic disorder is caused by the presence of an extra copy of all or part of chromosome 21 (*Homo sapiens* autosome 21, or HSA21) (Antonarakis et al, 2004; Antonarakis, 2017). This chromosome carries 235 protein-coding genes and 441 non–protein-coding genes (Ensembl release 106 – April 2022). The possibility of triplication for such a large number of genes makes DS one of the most complex genetic conditions compatible with viability. DS is associated with a broad spectrum of clinical symptoms, but the features common to all DS variants include an intellectual deficit that impairs learning and memory and an increase in the risk of developing a form of dementia resembling Alzheimer's disease (AD), even in patients as young as 40 yr of age (Dierssen, 2012; Wiseman et al, 2015; Ballard et al, 2016). The precise contribution of the overexpression of each HSA21 gene to the cognitive impairment observed in DS remains unknown.

We studied two DS mouse models. The first was the *Dyrk1A* BAC 189N3 model, carrying a triplication of the ~152-kb mouse *Dyrk1a*

[1]Centre Psychiatrie and Neurosciences, INSERM U894, Paris, France [2]Laboratoire de Génomique Fonctionnelle, CNG, Commissariat à l'Énergie Atomique et aux Énergies Alternatives (CEA), Evry, France [3]Instituto de Neurociencias, Consejo Superior de Investigaciones Científicas-Universidad Miguel Hernández (CSIC-UMH), Universidad Miguel Hernandez-Campus de San Juan, San Juan, Spain [4]Department of Neuroscience, The Johns Hopkins University School of Medicine, Baltimore, MD, USA [5]Department of Biological Sciences, Korea Advanced Institute of Science and Technology (KAIST), Center for Synaptic Brain Dysfunctions, Institute for Basic Science (IBS), Daejeon, Republic of Korea [6]Department of Molecular Biology, Division of Biological Science, Nagoya University Graduate School of Science, Nagoya, Japan [7]Department of Biological Sciences, University of Toledo, Toledo, OH, USA [8]Department of Molecular Pharmacology and Cell Biology, Leibniz Institut für Molekulare Pharmakologie (FMP) and Freie Universität Berlin, Berlin, Germany [9]Hybrigenics, Paris, France [10]Department of Cellular and Molecular Biology, Roswell Park Division of Graduate School, State University of New York at Buffalo, Buffalo, NY, USA [11]Institut Jérôme Lejeune, Paris, France [12]Institut de Génétique et de Biologie Moléculaire et Cellulaire, Illkirch, France [13]Centre National de la Recherche Scientifique (CNRS), UMR7104, Illkirch, France [14]INSERM, U964, Illkirch, France [15]Université de Strasbourg, Illkirch, France [16]PHENOMIN, Institut Clinique de la Souris, ICS, GIE CERBM, CNRS, INSERM, Université de Strasbourg, Illkirch-Graffenstaden, France [17]Université Paris-Saclay, CNRS, ENS Paris-Saclay, CentraleSupélec, LuMIn, Gif sur Yvette, France [18]Department of Biology, Ecole Normale Supérieure Paris-Saclay Université Paris-Saclay, Gif sur Yvette, France

Correspondence: michel.simonneau@ens-paris-saclay.fr
*Julia Viard and Yann Loe-Mie are co-authors.
**Michel Simonneau and Aude-Marie Lepagnol-Bestel are senior co-authors.

locus containing the entire mouse *Dyrk1a* (dual-specificity tyrosine phosphorylated and regulated kinase 1A) gene together with a 6 kb flanking fragment on the 5′ side and a 19 kb flanking fragment on the 3′ side (Guedj et al, 2012). The second model was a transgenic mouse line (Dp(16)1Yey) carrying a triplication of ~23.3 Mb from *Mus musculus* chr16 (Mmu16) syntenic to 115 coding genes from HSA21 (Li et al, 2007; Aziz et al, 2018) including *DYRK1A*, precisely reflecting the gene dosage of HSA21 orthologs. The *Dyrk1A* gene has been shown to play a major role in DS; its overexpression induces changes to synaptic plasticity in both the hippocampus and pre-frontal cortex (Ahn et al, 2006; Thomazeau et al, 2014; Atas-Ozcan et al, 2021). Dyrk1a is an important candidate protein for involvement in the learning and memory impairment seen in DS patients (Smith et al, 1997), but the regulatory pathways impaired by *DYRK1A* trisomy have yet to be identified.

We investigated the respective contributions of Dyrk1a and other HSA21 gene products to the pathways underlying ID in DS. RNA-seq analysis on transcripts misregulated in the embryonic hippocampus revealed two contrasting gene repertoires: a repertoire of genes encoding chromatin-related proteins for the 189N3 *Dyrk1A* trisomy model, and a repertoire of genes encoding synapse-related proteins for the Dp(16)1Yey model. We then investigated the molecular network of proteins underlying DS phenotypes, by searching for human brain proteins interacting with proteins encoded by HSA21. To this end, we conducted a large-scale yeast two-hybrid screen with HSA21 baits and a human brain library of targets. This analysis revealed that both direct interactors of HSA21-encoded proteins and their direct rebounds are enriched in proteins involved in ID. We also found an enrichment in HSA21-encoded proteins within a protein network in the postsynaptic density of the dendritic spine. The same interactome was also found to be enriched in proteins involved in ARC-related synaptic plasticity, ASD, and late-onset Alzheimer's disease (LOAD).

# Results

### Whole-genome RNA sequencing reveals two contrasting networks of deregulated genes in the hippocampus for the 189N3 *DYRK1A* and Dp(16)1Yey DS models

We used the 189N3 and the Dp(16)1Yey/+ mouse models of DS. We performed RNA-seq analysis on E17 hippocampi for these two DS models, to identify differentially expressed genes (DEGs) relative to wild-type E17 hippocampi. We identified 84 DEGs (50 down-regulated and 34 up-regulated) in 189N3 mice (Table S1) and 142 DEGs (77 down-regulated and 65 up-regulated) in Dp(16)1Yey/+ mice (Table S2) relative to their wild-type littermate controls, with a false discovery rate < 0.05. Note that Dyrk1a is overexpressed in our 189N3 samples. Furthermore, 10 genes (of 65 up-regulated genes) located in the mouse chromosome 16 syntenic region (see Fig S1 in Aziz et al [2018]) are overexpressed in our Dp(16)1Yey/+ samples, including Robo2 (Table S2). Robo2 encodes an axon guidance receptor that is also involved in establishing synaptic specificity (Blockus et al, 2021).

Various tools were used for the analysis of DEGs: Amigo2 gene ontology (GO) analysis, String Protein–Protein Interaction (PPI)

Networks Functional Enrichment Analysis, Webgestalt, Suite (Liao et al, 2019), and SynGO—Synaptic Gene Ontologies and annotations (Koopmans et al, 2019).

Amigo2 identified GO categories for the down-regulated DEGs, with a deregulation of the expression of genes encoding chromatin-related proteins in 189N3 mice with:

GO:0006334~nucleosome assembly ($P$-value = 1.17 × 10$^{-8}$)
GO:0031497~chromatin assembly ($P$-value = 5.98 × 10$^{-8}$)
GO:0034728~ nucleosome organization ($P$-value = 1.71 × 10$^{-7}$).

This deregulation of genes encoding chromatin-related proteins is consistent with our previously reported data (Lepagnol-Bestel et al, 2009). In contrast, no significant GO categories were identified for the up-regulated DEGs.

In Dp(16)1Yey/+ mice, genes encoding proteins involved in synaptic function were found to be deregulated:

GO:0007268~chemical synaptic transmission ($P$-value = 6.87 × 10$^{-9}$)
GO:0051932~synaptic transmission, GABAergic ($P$-value = 1.27 × 10$^{-5}$)
GO:0048812~neuron projection morphogenesis ($P$-value = 8.24 × 10$^{-5}$)

For GO:0051932~synaptic transmission, GABAergic, 6 of 77 genes were deregulated for a repertoire of 53 of 24,850 human genes (7.8%), indicating a 36.54-fold enrichment relative to expectations (hypergeometric $P$-value = 1.48 × 10$^{-8}$). This result is entirely consistent with the impaired excitation-inhibition balance (E–I balance) of synaptic activity in DS mouse models (Kleschevnikov et al, 2012; Raveau et al, 2018).

We used String analysis to identify a gene network with a PPI enrichment $P$-value: <1.0 × 10$^{-16}$ that includes a nucleosome-related network for the down-regulated DEGs of 189N3 mice (Fig 1A). Interestingly, human homologs of *Hist1h1a* (*HIST1H1A*) and *Hist1h3a* (*HIST1H3A*) that are part of this nucleosome-related network are included in a locus associated to bipolar disorder and schizophrenia, according recent Genome Wide Association Studies (Mullins et al, 2021; Trubetskoy et al, 2022). String analysis identified a gene network with a PPI enrichment $P$-value: <1.0 × 10$^{-16}$ that includes a synapse-related network for the down-regulated DEGs of Dp(16)1Yey/+ mice (Fig 1B).

Using the curated ontology of the SynGO—Synaptic Gene Ontologies and annotations (Koopmans et al, 2019), we further examined the synaptic signal and found. We analyzed the 77 down-regulated DEGs. 25 of 77 DEGs were mapped to 25 unique SynGO annotated genes. The enriched cellular component ontology terms are: Synapse (n = 23) $P$ = 9.42 × 10$^{-11}$, Presynapse (n = 14) $P$ = 6.22 × 10$^{-8}$ and Postsynapse (n = 12) $P$ = 1.38 × 10$^{-5}$ (Fig 2).

A Dapple analysis (Rossin et al, 2011) of 189N3 and Dp(16)1Yey/+ DEGs identified two statistically significant contrasting networks (direct edge counts; $P$ < 0.05), revealing that the genetic variation induced by mmu16 triplication affects a limited set of underlying mechanisms (Fig S1). We then applied the Webgestalt suite (Liao et al, 2019) to the 77 genes down-regulated in Dp(16)1Yey/+ mice. We identified two significant networks: (A) a network consisting of eight of the 70 up-regulated genes, displaying an enrichment in the GO Biological Process chemical synaptic transmission ($P$ = 220446 × 10$^{-16}$); (B) a network of 4 of the 77 down-regulated genes, displaying enrichment in the GO Biological Process glutamate receptor

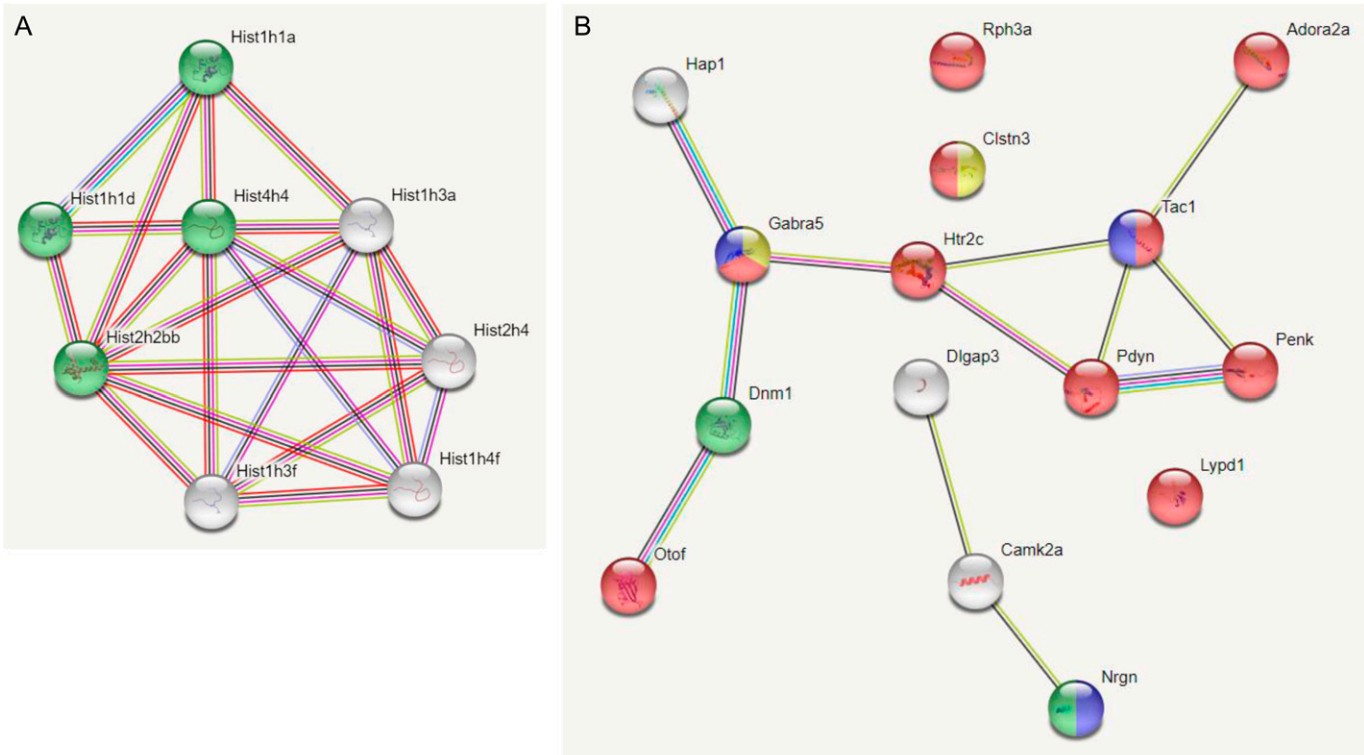

**Figure 1.  STRING Protein–Protein Interaction (PPI) Networks Functional Enrichment for proteins encoded by deregulated genes identified in E17 hippocampus of 189N3 and Dp(16)1Yey transgenic mouse models, respectively.**
We performed RNA sequencing on embryonic E17 hippocampi of these two DS models. We identified 84 deregulated genes (50 down-regulated and 34 up-regulated) in 189N3 (Table S1) and 142 deregulated genes (77 down-regulated; 65 up-regulated) in Dp(16)1Yey/+ (Table S2) compared with their littermate controls. **(A)** For the 50 down-regulated differentially expressed genes of 189N3 mice, we found a PPI enrichment $P$-value < $1.0 \times 10^{-16}$ with an enrichment in Gene ontology (GO) GO: 0006334~nucleosome assembly including Hist1h1a, Hist1h1d, Hist4h4, and Hist2h2bb (in green in the PPI network). **(B)** For the 77 down-regulated differentially expressed genes of Dp(16)1Yey/+, we found a PPI enrichment $P$-value = $5.83 \times 10^{-10}$. Are overrepresented in the PPI network: In red, 10 genes (Rph3a, Adora2a, Clstn3, Gabra5, Htr2c, Tac1n Pdyn, Penk, Lypd1, and Otof) GO:0007268 Chemical synaptic transmission ($P = 2.77 \times 10^{-11}$). In blue, three genes (Gabra5, Tac1, and Nrgn) GO:0008305 Associative learning ($P = 0.00173$). In yellow, two genes (Gabra5 and Clstn3) GO:0051932 Synaptic transmission, Gabaergic ($P = 0.0210$). In green: two genes (Dnm1; Nrgn) GO: 0044327 Dendritic spine head ($P = 0.0034$).

signaling pathway ($P = 220446 \times 10^{-16}$). Three genes (*Camk2a*, *Gda*, and *Dlgap3*) were part of a 20-protein network of ARC-dependent DLG4 interactors (Fernández et al, 2017), corresponding to a 43.85-fold enrichment relative to expectations (hypergeometric $P$-value = $4.20 \times 10^{-5}$) (parameters: 3, 20, 77, and 22,508 mouse genes from Mouse Ensembl [GRCm38.p6]) (Fig S2).

These results indicate the contrasting deregulation of a chromatin-related network for the 189N3 model and a synaptic plasticity-related network with an enrichment in genes linked to ARC postsynapse complexes involved in neural dysfunction and intelligence for the Dp(16)1Yey/+ model.

### Establishment of a HSA21 PPI map by high-throughput yeast two-hybrid (Y2H) screening: enrichment in ID genes

We performed a large-scale PPI study to improve our knowledge of the molecular network underlying DS. We performed 82 screens—72 with HSA21 protein baits and 10 screens against their direct interactors (rebounds) (Table S3)—with a highly complex random-primed human adult brain cDNA library. These interactions were ranked by category (a–f), with a Predicted Biological Score (PBS).

PBS is computed as an $e$-value and thresholds are attributed to define categories from high confidence (A) to lower confidence (D) interactions. The PBS $e$-value ranges from 0 to 1 and has been classified in five distinct categories: a to e. Inter-category thresholds were chosen manually with respect to a training data set containing known true-positive and false-positive interactions: a < $1 \times 10^{-10}$ < b < $1 \times 10^{-5}$ < c < $1 \times 10^{-2.5}$ < d < 1. Complete statistical analysis of the interactome leads to the identification of highly connected interacting domain for which the corresponding PBS has been set to 1. PBS f also set to one are experimentally validated false positive (interaction with the DNA binding domain [DBD]) (Formstecher et al, 2005).

An analysis of direct interactors from 72 HSA21 bait screens yielded 1,687 novel interactions, with the confirmation of 76 already known (Biogrid) interactions (Fig 3A and B). An analysis of direct interactors from 82 direct and rebound screens yielded 1,949 novel interactions, with the confirmation of 100 already known (Biogrid) interactions (Fig 3C and D). We then compared these direct interactors with three lists of genes involved in intellectual disability (ID), the S10 list ($n = 527$), the S11 list ($n = 628$) from reference Gilissen et al (2014), and the S2 list ($n = 1,244$) from reference

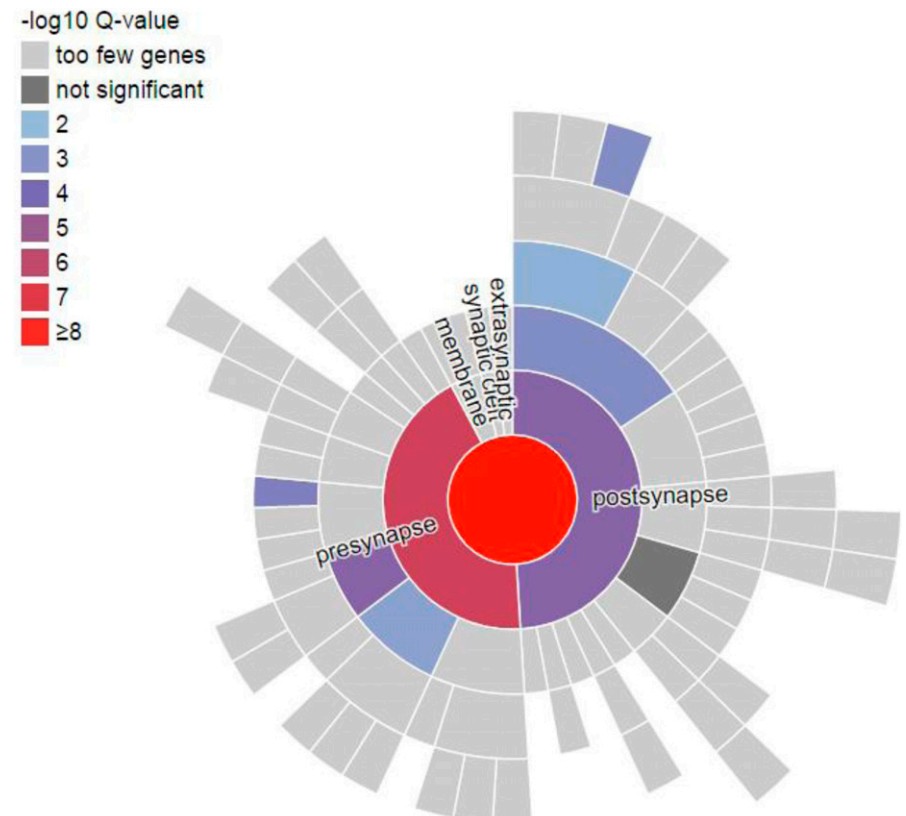

-log10 Q-value

| | |
|---|---|
| ☐ | too few genes |
| ■ | not significant |
| ☐ | 2 |
| ☐ | 3 |
| ☐ | 4 |
| ☐ | 5 |
| ☐ | 6 |
| ☐ | 7 |
| ☐ | ≥8 |

**Figure 2. SynGO analysis of differentially expressed genes (DEGs) identified in E17 hippocampus of Dp(16)1Yey transgenic mouse model.**

SynGO, Synaptic Gene Ontologies, is an evidence-based, expert-curated resource for synapse function and gene enrichment studies. We identified 142 and DEGs (77 down-regulated and 65 up-regulated) in Dp(16)1Yey/+ (Table S2) compared with their littermate wild-type controls, using a False Discovery Rate < 0.05. We analyzed the 77 down-regulated DEGs using SynGO. 25 of 77 DEGs were mapped to 25 unique SynGO annotated genes. The enriched cellular component ontology terms are: Synapse ($n = 23$) $P = 9.42 \times 10^{-11}$. Presynapse ($n = 14$) $P = 6.22 \times 10^{-8}$. Postsynapse ($n = 12$) $P = 1.38 \times 10^{-5}$. Neural dense core vesicle ($n = 4$) $P = 1.60 \times 10^{-5}$. Integral component of presynaptic membrane ($n = 5$) $P = 1.71 \times 10^{-4}$. Postsynaptic specialization ($n = 7$) $P = 4.04 \times 10^{-4}$. Integral component of presynaptic membrane ($n = 4$) $P = 4.92 \times 10^{-4}$. Synaptic vesicle ($n = 4$) $P = 1.87 \times 10^{-3}$. Postsynaptic density ($n = 5$) $P = 4.69 \times 10^{-3}$.

Deciphering Developmental Disorders Study (2015). Table S4 indicates genes found in both ID lists and interactor lists. HSA21 direct interactors are enriched in ID proteins (HSA21 bait direct interactors against S10: *P*-value = $2.29 \times 10^{-8}$; HSA21 bait direct interactors against S11: *P*-value = $9.39 \times 10^{-12}$; HSA21 bait direct interactors against S2: *P*-value = $7.53 \times 10^{-13}$) (Fig 3E). Similarly, HSA21 bait and rebound direct interactors were also enriched in ID proteins (HSA21 bait and rebound direct interactors against S10: *P*-value = $8.30 \times 10^{-9}$; HSA21 bait and rebound direct interactors against S11: *P*-value = $8.64 \times 10^{-12}$; HSA21 bait and rebound direct interactors against S2: *P*-value = $7.76 \times 10^{-14}$) (Fig 3F). Thus, both HSA21 direct interactors and rebound direct interactors are part of a large ID network.

We performed a biological process analysis with GO DAVID (see the Materials and Methods section) on direct interactors of both HDA21 baits and their rebounds (Fig 4). The colored nodes correspond to the most significant results: GO:0022008~Neurogenesis (*P*-value = $3.06 \times 10^{-17}$); GO:0048812~Neuron projection morphogenesis (*P*-value = $2.91 \times 10^{-13}$); GO:0050767~Regulation of neurogenesis (*P*-value = $2.66 \times 10^{-6}$); GO:0043632~Modification-dependent macromolecule catabolic process (*P*-value = $6.46 \times 10^{-5}$); GO:0051962~Positive regulation of nervous system development (*P*-value = $6.29 \times 10^{-6}$); GO:0045665~Negative regulation of neuron differentiation (*P*-value = $1.55 \times 10^{-7}$) (Table S5). Overall, our data indicate an enrichment in interactions related to neuronal differentiation.

## Linking HSA21 proteins to neurodevelopmental diseases, neuropsychiatric diseases, and LOAD: STX1A–DYRK1A, LIMK1–HUNK, DYRK1A-EP300, DYRK1A-CREBBP, DYRK1A-FAM53C, DYRK1A-RNASEN, and DYRK1A-CLU

We focused our analysis on interactions of potential importance in brain diseases, by combining Y2H interaction data with proximity ligation assays (PLAs), which can localize PPIs at the subcellular level if the maximal distance between the antibodies required to generate a signal is 40 nm (Söderberg et al, 2006).

We first studied two novel interactions: STX1–DYRK1A and LIMK1–HUNK with *STX1* and *LIMK1* genes involved in Williams syndrome (WS) (Fig S3A). WS is a relatively rare microdeletion disorder affecting 1:7,500 individuals. It is caused by a hemizygous deletion of ~1.6 megabases and leads to the loss of one copy of 25–27 genes on chromosome 7q11.23. This deletion results in a unique disorder that affects multiple systems and is characterized by a specific cognitive and behavioral profile, including ID and hypersociability (Meyer-Lindenberg, 2009; Kozel et al, 2021).

*STX1A* and *LIMK1* are among the candidate genes underlying this specific cognitive and behavioral profile. STX1A, a neuronal regulator of presynaptic vesicle release (Südhof, 2014), may be involved in the cognitive profile of WS patients and may be a component of the cellular pathway determining human intelligence (Gao et al, 2010). Hemizygosity for the LIMK1 (LIM-kinase1) gene has

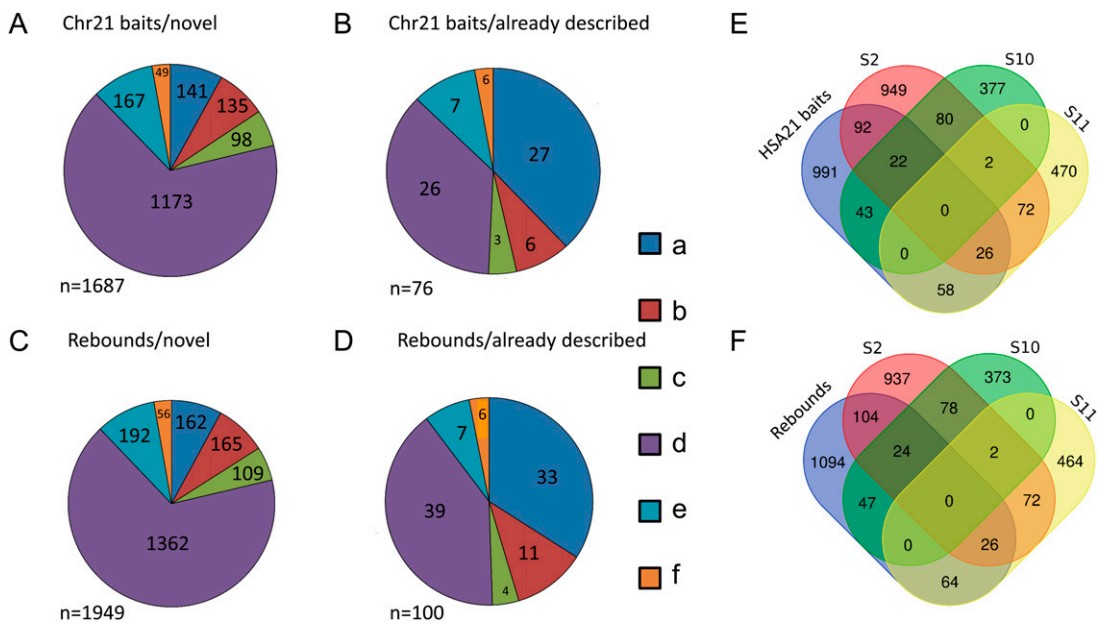

**Figure 3. High-throughput Y2H identifies 3,636 novel direct interactions with their enrichment in proteins involved in Intellectual Disabilities.**
72 screens with HSA21 protein as baits and 82 screens against their direct interactors (rebounds) have been performed using a human brain library. 1,687 and 1,949 novel direct interactions have been identified. These interactions have been ranked by category (a–f), using a Predicted Biological Score (Formstecher et al, 2005). **(A, B, C)** Analysis of direct interactors from 72 HSA21 baits screens (A, B, C). **(A, B)** 1,687 novel interactions were identified (A) and 76 already known (Biogrid) interactions confirmed (B). **(D, E, F)** Analysis of direct interactors from 82 rebound screens (D, E, F). **(D, E)** 1,949 novel interactions were identified (D) and 100 already known (Biogrid) interactions confirmed (E). We compared these direct interactors with three lists of genes involved in Intellectual Disability (Gilissen et al, 2014; Deciphering Developmental Disorders Study, 2015). **(C, F)** Both HSA21 direct interactors (C) and rebound direct interactors (F) are enriched in ID proteins (see text) suggesting that these two types of interactors are part of a large ID network.

been implicated in impaired visuospatial constructive cognition (Frangiskakis et al, 1996). Our Y2H study provided evidence of STX1A-DYRK1A and LIMK1-HUNK interactions (*DYRK1A* and *HUNK* being HSA21 genes). PLA showed that these STX1A–DYRK1A and LIMK1–HUNK interactions occurred in the dendrite (Fig S3B–E).

We then studied three interactions identified in a large assay using non-neuronal cells (Varjosalo et al, 2013): DYRK1A-EP300, DYRK1A-CREBBP, and DYRK1A-FAM53C (Fig S4A and B). Using immunoprecipitation of EP300 or CREBBP in HEK293, we identified DYRK1A in the immunoprecipitates (Fig S4C). Interestingly, EP300 and CREBBP are key proteins involved in the late phase of the long-term potentiation (L-LTP) of the synapse (Fig S5). Mouse models of the Rubinstein-Taybi syndrome (RTS), an inheritable disorder caused by mutations in the gene encoding the CREB binding protein (CREBBP) display impairment of some forms of long-term memory, and the late phase of hippocampal long-term potentiation (L-LTP) (Alarcón et al, 2004).

We identified DYRK1A–FAM53C interaction in human brain (Fig S4A). We detected interactions in both directions, with FAM53C interacting with the DYRK1A (UniProtKB - Q13627; DYR1A_HUMAN) kinase domain (128 AA–402 AA). The interaction was validated in hippocampal neurons using PLA (Fig S4D and E). Interestingly, an SNP within the FAM53C-KDM4 locus (Fig S6) has been reported to be associated with ASDs (Autism Spectrum Disorders Working Group of The Psychiatric Genomics Consortium, 2017). Fam53c-knockout mouse phenotypes found in (Fam53c<em1(IMPC)J>/Fam53c<em1(IMPC)J) mice include abnormal behavior with poor exploration

of new environments and low levels of thigmotaxis (an altered emotional response related to the anticipation of a nonspecific threat).

Finally, we studied two novel interactions: the interaction between the HSA21 *DSCR9* gene product and CLU, a risk factor for LOAD, and that of the HSA21 *DYRK1A* gene product with DROSHA/RNASEN, a microprocessor complex subunit (Fig 5A). As no bona fide antibody against DSCR9 was available, we generated a GFP-DSCR9 construct for DSCR9 protein imaging. PLA showed that the interaction between DSCR9 and CLU occurred in the nucleus. In situ PLA with anti-GFP and anti-CLU antibodies detected the interaction between these two proteins within the nuclei of primary cortical neurons (Fig 5B). *DSCR9* and *DSCR10* have been identified as genes found exclusively in primates, such as chimpanzee, gorilla, orangutan, crab-eating monkey, and African green monkey; they are not found in non-primate mammals (Takamatsu et al, 2002). The CLU gene has been identified as one of the top 20 genetic risks for LOAD (Lambert et al, 2013), and this finding was confirmed in a recent meta-analysis of genome-wide association studies (GWAS) on clinically diagnosed LOAD (94,437 individuals) (Kunkle et al, 2019). Our results indicate a direct nuclear interaction between the product of an HSA21 gene contributing to the genomic basis of the uniqueness of the primate phenotype and a LOAD risk gene. For the DYRK1A-DGRC8 interaction, we first validated this interaction by immunoprecipitation in native conditions (no overexpression) in HEK293 cells (Figs 5C and S7). This interaction was localized to neuronal nuclei by PLA with anti-GFP and anti-Rnasen antibodies

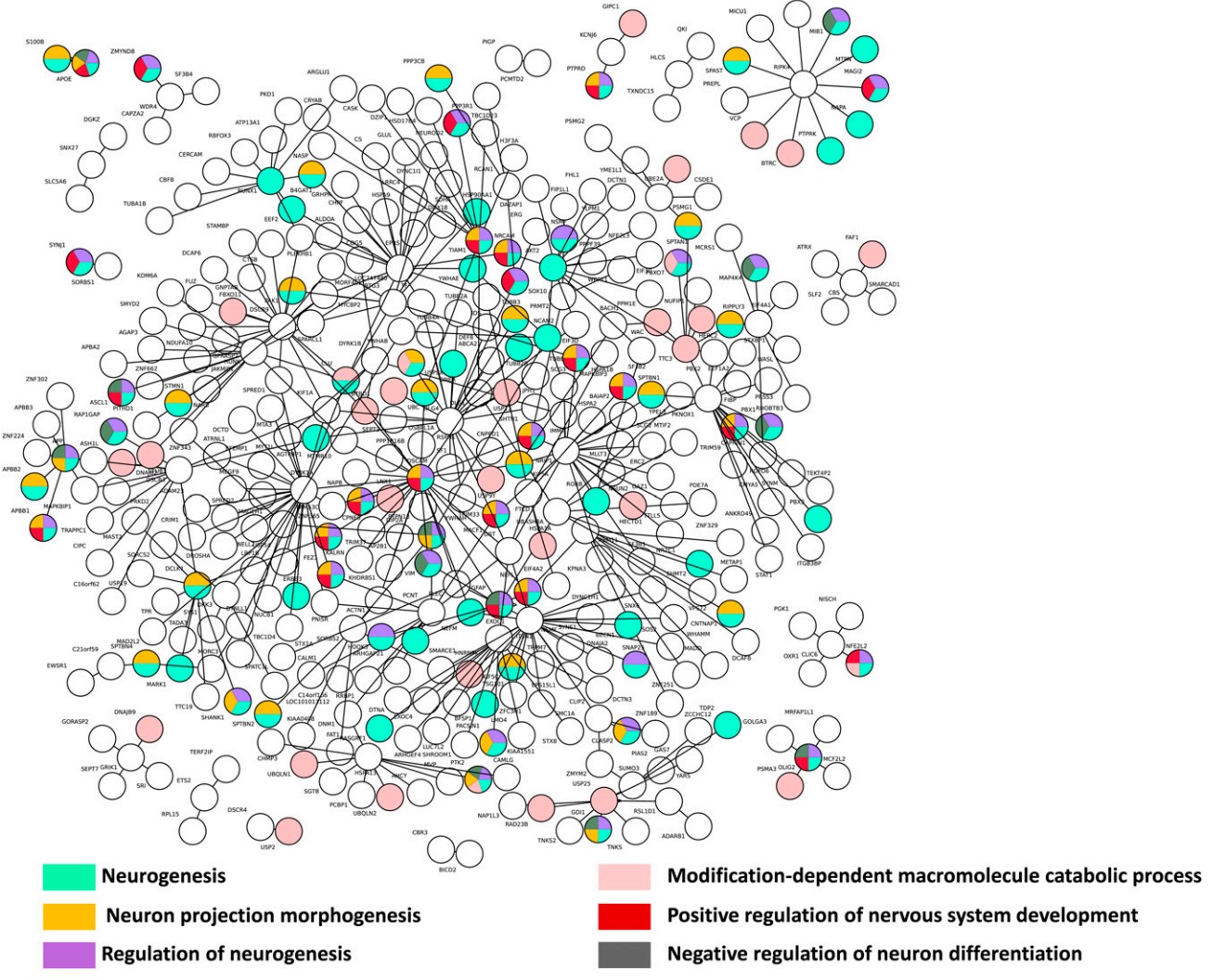

**Figure 4. Biological processes network interactions from Yeast two-hybrid protein–protein interaction data.**
A biological processes analysis using gene ontology (GO) DAVID was realized (see the Materials and Methods section). The colored nodes correspond to the most significant results: GO:0022008~Neurogenesis; GO:0048812~Neuron projection morphogenesis; GO:0050767~Regulation of neurogenesis; GO:0043632~Modification-dependent macromolecule catabolic process; GO:0051962~Positive regulation of nervous system development; GO:0045665~Negative regulation of neuron differentiation with $P$-value $3.06 \times 10^{-17}$, $2.91 \times 10^{-13}$, $2.66 \times 10^{-6}$, $6.46 \times 10^{-5}$, $6.29 \times 10^{-6}$, $1.55 \times 10^{-7}$, respectively. A color corresponds to a cluster of several biological processes. The multi-colored nodes correspond to genes presents in different annotation clusters.

on primary cortical neurons transfected on day 5 in culture (DIC5) with a Dyk1a–GFP construct (Fig 5D).

The microprocessor complex is a protein complex involved in the early stages of miRNA processing in animal cells. The minimal form of this complex consists of the ribonuclease enzyme DROSHA/RNASEN and the RNA-binding protein DGCR8 (also known as PA-SHA); it cleaves primary miRNA substrates to generate pre-miRNA in the cell nucleus (Wilson & Doudna, 2013).

A deficiency of *Dgcr8*, a gene disrupted by the 22q11.2 microdeletion responsible for schizophrenia in humans, alters short-term plasticity in the prefrontal cortex (Fénelon et al, 2011). *DYRK1A* overexpression would be expected to affect the function of the DYRK1A–RNASEN–DGRC8 interactome. Changes to the miRNA network can cause neurodegenerative disease (Hébert & De Strooper, 2009). Our results therefore suggest that the DYRK1A–RNASEN

interaction may be of direct relevance for understanding early AD in individuals with DS.

### Interactome of HSA21 proteins located in the dendrite: enrichment in an ARC-related protein network, high-risk genes for ASD, and LOAD risk factors

We performed PLA to validate the PPIs identified by Y2H and to determine the subcellular location of interactions. Our working hypothesis was that a subset of HSA21 proteins and their interactors might localize to the dendritic spine.

We were able to detect 21 PPIs in the dendrite, 20 of which were novel (GRIK1-HCN1; GRIK1-KCNQ2; GRIK1-SEPT7; GRIK1-KALRN; GRIK1-DLG4; HUNK-AGAP3; HUNK-SYNPO; HUNK-LIMK1; TIAM1-BIN1; TIAM1-DLG1; KCNJ6-DLG1; KCNJ6-DLG4; KCNJ6-DLG2; ITSN1-SNAP25;

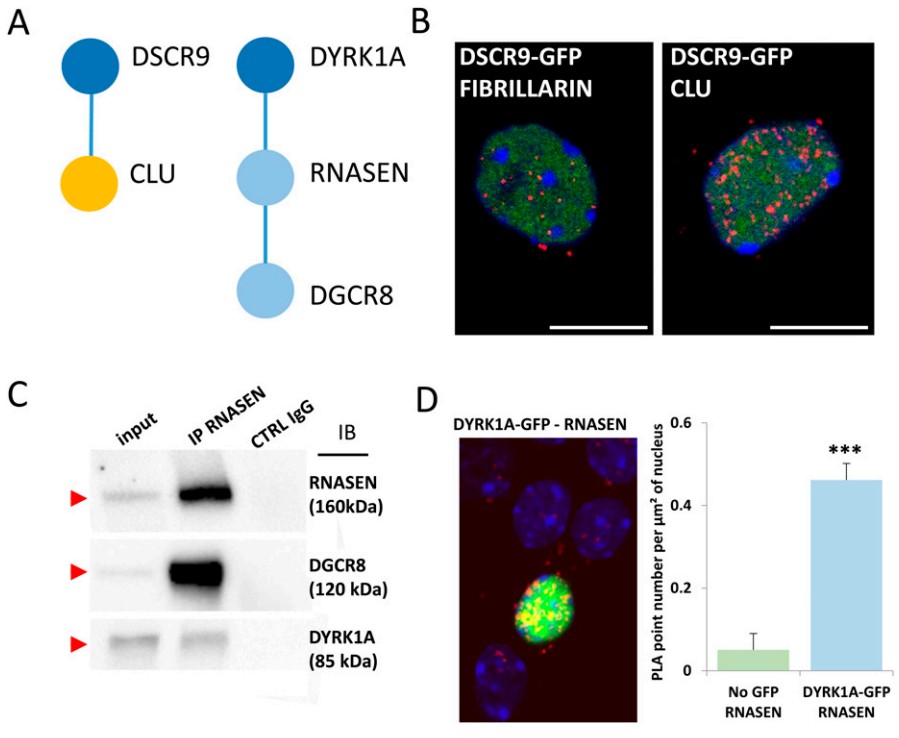

**Figure 5. Interactions of HSA21 proteins with proteins involved in late-onset Alzheimer's disease, intellectual disability, and neuropsychiatric diseases.**
**(A)**. Schematic representation of protein–protein interactions identified by yeast two-hybrid using a human brain library. Dark blue circles indicate HSA21-encoded proteins; orange circle indicates a late-onset Alzheimer's disease-related protein. **(B)**. In situ proximity ligation assay (PLA) on primary cortical neurons transfected at DIC5 and fixed 48 h later at DIC7 (red fluorescence) using anti-GFP and anti-Clu antibodies. PLA using anti-GFP and anti-Fibrillarin antibodies were performed as a negative control. Green fluorescent protein was visualized on green channel and heterochromatin was labelled using Topro3 (blue fluorescence). **(C, D)** DYRK1A interaction with DROSHA/RNASEN. **(C)** HEK293 cells were immunoprecipitated (IP) using anti-RNASEN antibody and anti-IgG antibody as a negative control. The input and precipitated fractions were then resolved by sodium dodecyl sulphate–polyacrylamide gel electrophoresis (SDS–PAGE) and analyzed by Western blot using anti-Rnasen, anti-Dgcr8, and anti-Dyrk1a antibodies. The arrows indicate protein bands at the expected size. Note that no cross-reaction was found with the IgGs. **(D)** In situ PLAs on primary cortical neurons transfected at DIC5 with Dyrk1a-GFP construct (green fluorescence) and fixed at DIC7, using anti-GFP and anti-Rnasen antibodies (red fluorescence). Non-transfected neurons were used as a negative control. Nuclei were labelled using Toprol staining (blue fluorescence). Mean interaction point numbers were calculated in heterochromatin of at least 25 transfected cortical neurons. ***P < 0.0005.

ITSN1-DLGAP1; DSCAM-DLG4; DSCAM-DLG2; SIPA1L1-DLG4; DLG2-GRIN2A; DLG2-GRIN2B), the only interaction having already been documented in BioGrid but not validated in the dendrite is SIPA1L1-DYRK1A (Fig S9). Antibodies used in the study are indicated in the Table S6.

We first focused on PPIs involving the HSA21 gene product GRIK1 (Fig 6A). This protein is one of the GRIK subunits known to function as a ligand-gated ion channel. Kainate receptors (KARs) are ubiquitous in the central nervous system, in both pre- and post-synaptic positions (Lerma & Marques, 2013). We first investigated the GRIK1–KCNQ2 interaction. KCNQ2 potassium channels are known to interact functionally with HCN1 potassium channels in the dendritic spines of the prefrontal cortex (Arnsten et al, 2012). We therefore first used PLA to check that HCN1, KCNQ2, and GRIK1 interacted physically in dendritic shafts and spines. We observed a direct interaction between GRIK1 and HCN1 in these compartments. We also identified and validated interactions of GRIK1 with SEPT7 and KALRN. SEPT7, a member of the septin family of GTPases, localizes to the dendritic branching points and spine necks (Tada et al, 2007). KALRN is a Rho-GEF localizing exclusively to the post-synaptic side of excitatory synapses (Penzes & Jones, 2008). It binds the NMDA receptor subunit Nr2b (Grin2b) (Kiraly et al, 2011). These results suggest that GRIK1 is part of two synaptic complexes, one located near PSD-95 (DLG4) at the tip of the dendritic spine, and the other at the neck of the spines.

The Y2H screen showed that the HSA21 gene product HUNK (also known as MAK-V) interacted with the GTPase-activating protein AGAP3, the actin-associated protein synaptopodin (SYNPO) and the synapse-related LIMK1 protein. These three interactions were validated in dendrites by PLA (Fig 6B). AGAP3 was recently identified as an essential signaling component of the NMDA receptor complex linking NMDA receptor activation to AMPA receptor trafficking (Oku & Huganir, 2013). SYNPO was localized to the necks of dendritic spines and was linked to the spine apparatus, suggesting a key role in the regulation of synaptic plasticity (Korkotian et al, 2014). These results suggest that HUNK is involved in complexes localized both near PSD-95 and in the spine apparatus. LIMK1 is involved in in-tracellular signaling and is strongly expressed in the brain; it has been suggested that LIMK1 hemizygosity results in an impairment of visuospatial constructive cognition (Frangiskakis et al, 1996).

We then analyzed the interaction of the HSA21 gene product TIAM1 with BIN1 and DLG1. TIAM1 is a Rac1-associated GEF 1 involved in synaptic plasticity (Penzes & Rafalovich, 2012) and specifically expressed in subgroups of glutamatergic synapses, such as the dendritic spines of the perforant path-dentate gyrus hippocampal synapse (Rao et al, 2019). BIN1 was the second risk factor for LOAD, after APOE4, to be identified by GWAS (Lambert et al, 2013; Kunkle et al, 2019). BIN1 has multiple functions, including a postsynaptic role (Daudin et al, 2018 Preprint; Schürmann et al, 2020). PLA provided evidence for TIAM1–BIN1 and TIAM1–DLG1 interactions in dendrites (Fig 6B).

Another important set of interactions identified by the Y2H screen and validated by PLA (Fig 6C) were those between the HSA21-encoded potassium channel KCNJ6, a voltage-insensitive potassium channel from the kainate ionotropic glutamate receptor (GRIK) family, and three members of the DLG family: DLG1, DLG2, and DLG4. The number of KCNJ6–DLG2 interactions was larger in the Dp(16)1Yey transgenic mouse model than in the control, whereas

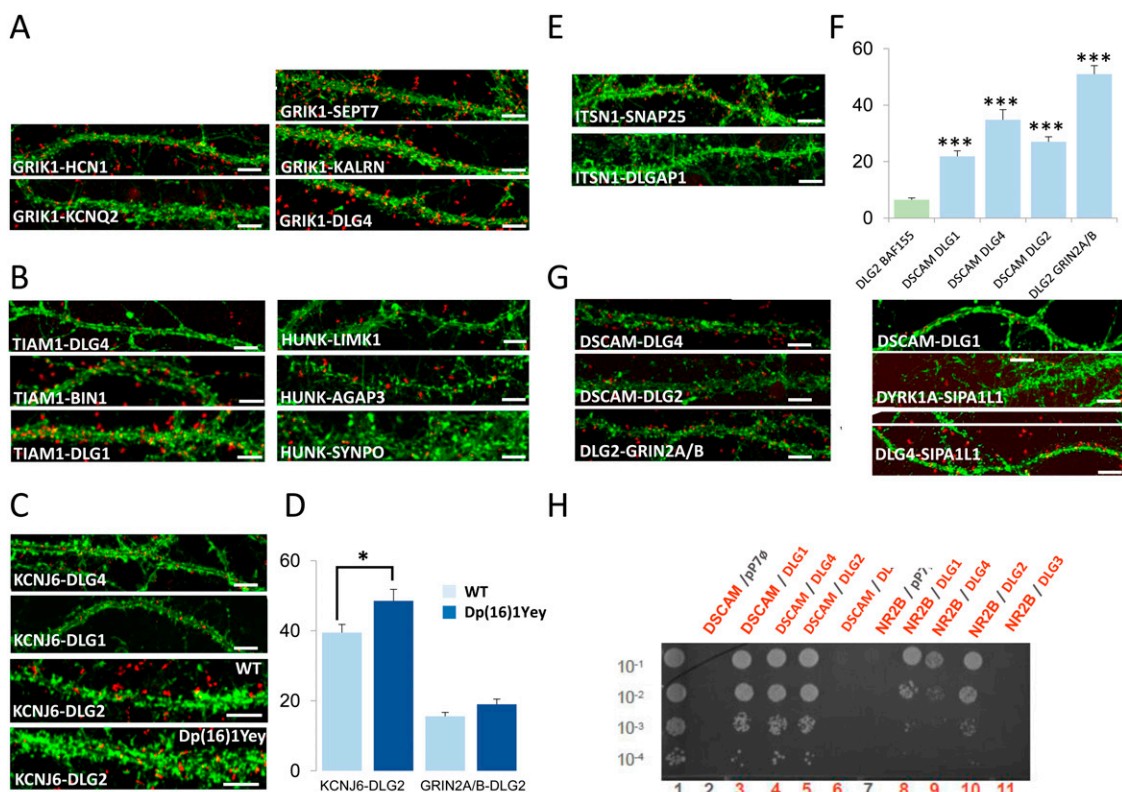

**Figure 6. Chr21-encoded proteins have direct interactors in dendritic spine PSD.**
**(A, B, C, D, E, F, G)**. In situ proximity ligation assays (PLA) on primary cortical neurons fixed at DIC21 (red fluorescence). Dendritic network and dendritic spines were labelled using phalloidin staining (green fluorescence). **(A)**. PLA of GRIK1 with direct interactors, HCN1, KCNQ2, SEPT7, KALRN, and DLG4. **(B)**. PLA of TIAM1 with direct interactors DLG4, BIN1, and DLG1. PLA of HUNK with LIMK1, AGAP3, and SYNPO. **(C, D)** In situ PLA on transgenic Dp(16)1Yey and WT primary cortical neurons fixed at DIC21 (red fluorescence) using anti-Dlg2 and anti-Kcnj6 or anti-Grin2ab antibodies. Quantification of interactions. Mean interaction point numbers were calculated in dendrites of at least 30 cortical neurons at DIC21 (from three different embryos per genotype). *$P < 0.05$ Scale bars = 10 $\mu$m. Mean interaction point numbers were calculated in dendrites of 25–30 cortical neurons at DIC21. **(E)**. PLA of ITSN1 with direct interactors SNAP25 and DLGAP1. **(F)** Quantification of interactions between DSCAM and DLG1, DLG2 or DLG4; Quantification of interactions between DLG2 and GRIN2A/B. **(G)** PLA of DSCAM with direct interactors DLG4, DLG1, and DLG2. PLA of DYRK1A with its direct interactor SIPA1L1, of DLG2 with GRIN2A/B and of DLG4 with SIPA1L1 as direct interactors. **(H)** Yeast two-hybrid one-by-one assays revealed DSCAM and NR2B as interactors of some of DLGs. Lane 1 is the positive control. Lanes 2 and 7 are the negative controls (pP7-DSCAM or pP7-NR2B vector with empty pP7 vector). Lanes 3–6 and 8–11 are the DSCAM and NR2B interactions, respectively. Please see Fig S8 for negative controls and Fig S9 for quantification of PLAs.

the number of GRIN2A/B–DLG2 interactions was unaffected (Fig 6D). *KCNJ6* is expressed in dendrites and dendritic spines, at the postsynaptic density (PSD) of excitatory synapses (Drake et al, 1997; Luján et al, 2009), and trisomy for this gene leads to synaptic and behavioral changes (Cooper et al, 2012).

Another two interesting interactions detected in the Y2H screen and validated by PLA (Fig 6E) were those between the HSA21 gene product intersectin (ITSN1), SNAP25, and DLG-associated protein 1 (DLGAP1/GKAP). SNAP25, a member of the SNARE protein family, is not only essential for the exocytosis of synaptic vesicles (Südhof & Rothman, 2009; Südhof, 2014), but also involved in the trafficking of postsynaptic NMDA receptors (Jurado et al, 2013) and spine morphogenesis (Tomasoni et al, 2013). DLGAP1 is a core protein of the scaffolding complex of the synapse (Kim et al, 1997). The detection of these interactions is consistent with the *Itsn1* mutant mouse phenotype, which is characterized by severe deficits of spatial learning and contextual fear memory (Sengar et al, 2013) and of synaptic hippocampal plasticity (Jakob et al, 2017).

We also localized the interactions of DSCAM with DLG2 (Discs large 2) and DLG4 identified by Y2H to the dendritic spines by PLA

(Fig 6F and G). Using DSCAM as bait against individual DLG family members (a one-by-one Y2H approach), we identified interactions between DSCAM and each of the four members of the DLG family: DLG1, DLG2, DLG3, and DLG4 (Fig 6H). DSCAM is known to regulate dendrite arborization and spine formation during cortical circuit development (Maynard & Stein, 2012). DLG1 (also known as SAP97), DLG2 (also known as PSD93/chapsyn-110), and DLG4 (also known as PSD-95/SAP90) are known to bind various proteins and signaling molecules at the PSD (Kim & Sheng, 2004; Sheng & Kim, 2011). Intriguingly, cognition is abnormal in both mice lacking *Dlg2* and humans with *Dlg2* mutations (Nithianantharajah et al, 2013).

SIPA1L1, also known as SPAR, is a Rap-specific GTPase-activating protein (RapGAP) that regulates actin dynamics and dendritic spine morphology, and is degraded by the ubiquitin-proteasome system (Pak et al, 2001; Pak & Sheng, 2003). From the Y2H results, we found evidence of direct interactions between SIPA1L1 and DYRK1A or DLG4 and between DYRK1A and DLG4.

Control interactions and quantification of PLA experiments are illustrated in Figs S8 and S9, respectively.

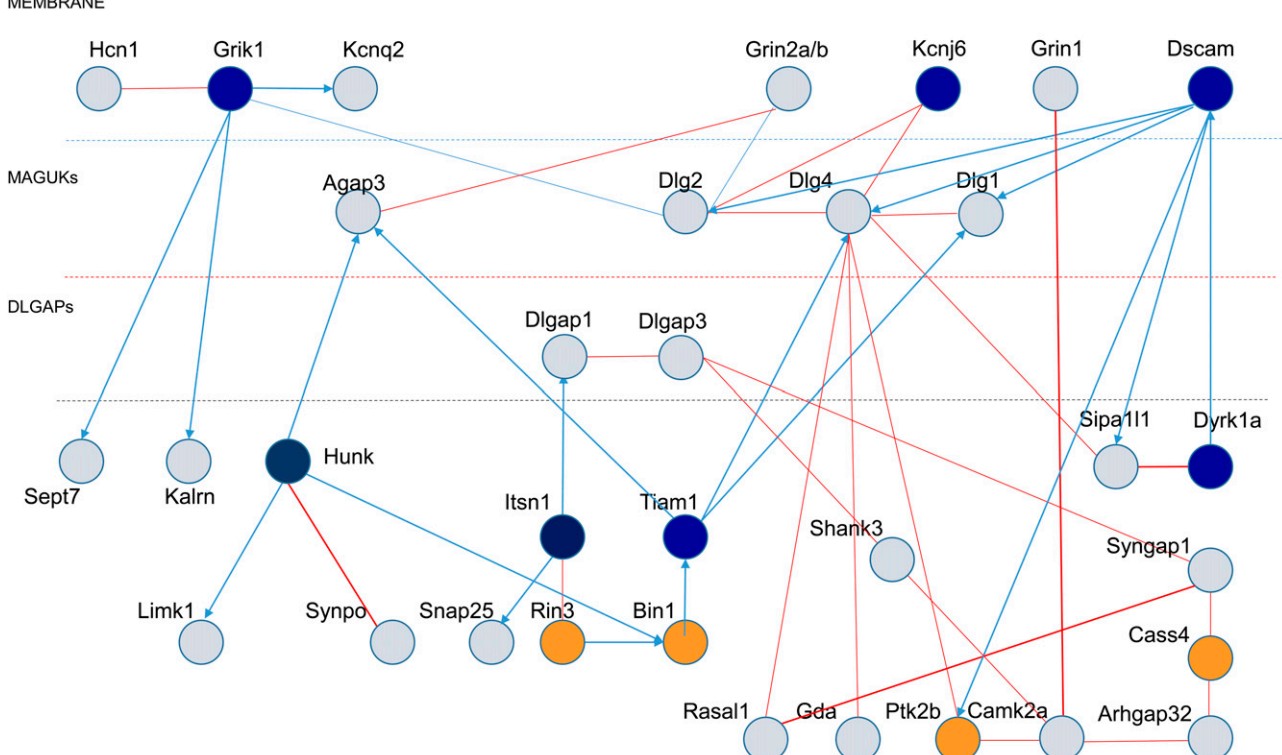

**Figure 7. Protein–protein interactions in the three layers of dendritic spine PSD: enrichment in proteins encoded by either HSA21 or late-onset Alzheimer's disease-GWAS genes.**
Schematic representation of synaptic protein–protein interactions performed by yeast-two-hybrid, with the three layers of dendritic spine PSDs indicated (membrane; MAGUKs and DLGAPs). HSA21-encoded proteins are represented as dark blue circles. Late-onset Alzheimer's disease-GWAS encoded proteins are represented by dark orange circles.

We defined a putative synaptic network of 33 proteins (Fig 7) based on the Y2H results, and its position in a four-layer model, as proposed by Li et al (2016, 2017). The four layers consist of (i) a membrane layer for ionic channels, neurotransmitter receptors and cell-adhesion molecules, (ii) a second layer for DLGs, (iii) a third layer for DLGAPs, and (iv) a fourth layer for direct DLGP interactors (such as SHANKs).

We analyzed the enrichment of the synaptic protein network of 33 products (Fig 7) in repertoires linked to human cognition.

We observed an enrichment in the ARC-dependent postsynaptic complex involved in neural dysfunction and intelligence (Fernández et al, 2017). We detected enrichment for 20 proteins in this group, 10 of which (Camk2a, Dlgap1, Dlgap3, Dlg1, Dlg2, Dlg4, Gda, Grin1, Grin2a, and Syngap1) are part of our synaptic complex (parameters: 10, 20, 33, and 20,471; over enriched 310.17-fold compared with expectations; hypergeometric $P$-value = $4.76 \times 10^{-24}$ [Ensembl release 106 - Apr 2022]).

The 33-protein network was also enriched in genes associated with a high risk of ASDs. DSCAM and DYRK1A were identified as part of this network, which also included DLGAP1 and SHANK3. These five genes are considered to confer an ~20-fold increase in risk, within a group of 26 genes (Sanders et al, 2015; Willsey et al, 2018; Satterstrom et al, 2020). In ASD, studies leveraging the statistical power afforded by rare de novo putatively damaging variants have identified more than 65 strongly associated genes (Sanders et al, 2015). The most deleterious variants (likely gene disrupting or LGD

variants) in the highest confidence subset of these genes ($N$ = 26), as a group, increase the risk by about 20-fold, and LGD variants in the highest-confidence genes within this subset carry even greater risks (Willsey et al, 2018; Satterstrom et al, 2020). We observed 119.29-fold enrichment relative to expectations (hypergeometric $P$-value = $5.09 \times 10^{-10}$; parameters: 5, 26, 33, and 20,471).

The third group of proteins for which enrichment was detected was the LOAD group, which included 11 new loci corresponding to 26 candidate genes (Lambert et al, 2013; Karch et al, 2014). We observed a 95.44-fold enrichment relative to expectations (hypergeometric $P$-value = $8.15 \times 10^{-8}$; parameters: 4, 33, 26, and 20,471).

These results demonstrate an enrichment of our postsynaptic network in HSA21 proteins, ASD high-risk gene products, proteins of the ARC-related protein network, and LOAD risk factors.

### DSCAM–DYRK1A interaction

In our Y2H screens, we identified interactions between human DSCAM, its human paralog DSCAML1 and human DYRK1A. Both DSCAM and DYRK1A belong to a subset of 26 genes conferring an ~20-fold increase in ASD risk (Sanders et al, 2015; Willsey et al, 2018; Satterstrom et al, 2020).

The Y2H screens used here made it possible to identify the domain of the prey interacting with the bait. Once positive clones had been identified, overlapping prey fragments derived from the

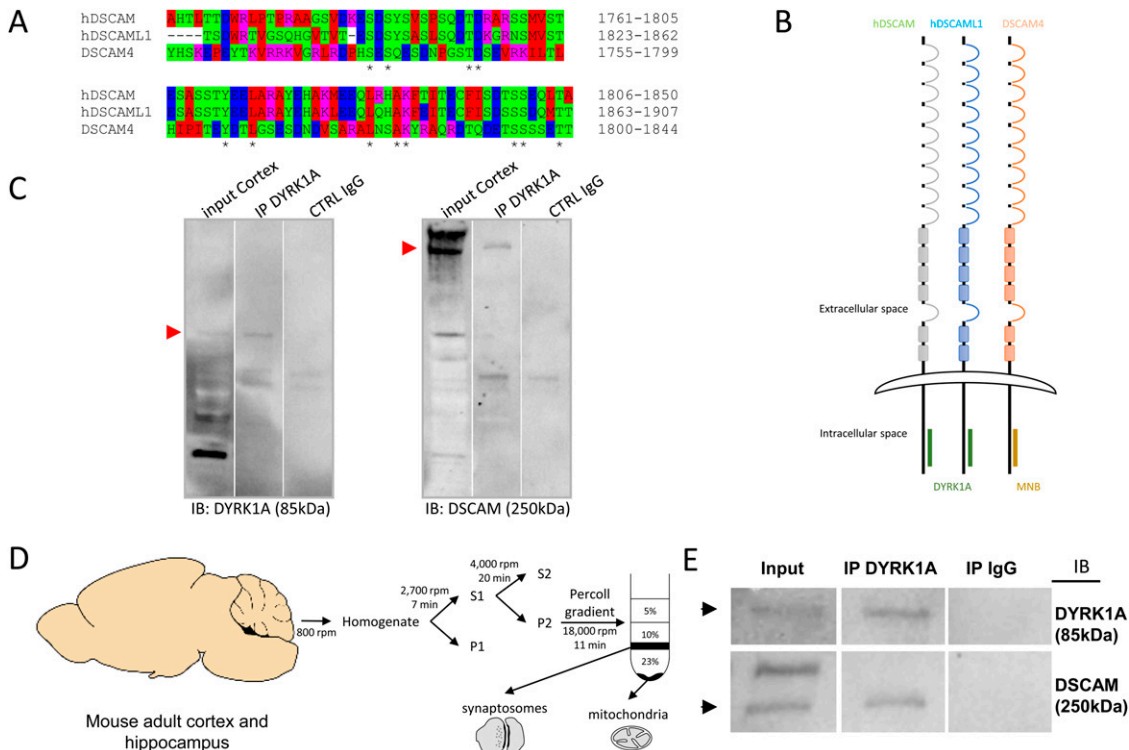

**Figure 8. Conservation of DSCAM–DYRK1A interaction in human and in drosophila.**
**(A)** AA alignment of the DSCAM domain that interacts with DYRK1A and Minibrain. This alignment was performed with ClustalW 2.1 software. **(B)** Schematic representation of DSCAM and DYRK1A protein family interaction. Human DSCAM (hDSCAM in green), human DSCAML1 (hDSCAML1 in blue) and its drosophila ortholog (dDSCAM4 in red) share the same conserved protein domain interacting with human DYRK1A (hDYRK1A) or its drosophila ortholog (MNB), respectively. **(C)** Adult mouse cortical protein extract were immunoprecipitated (IP) using anti-Dyrk1a antibody and anti-IgG antibody as a negative control. The input and precipitated fractions were then resolved by sodium dodecyl sulphate–polyacrylamide gel electrophoresis (SDS–PAGE) and analyzed by Western blot using anti-Dyrk1a and anti-Dscam antibody. Red arrows indicate protein bands at the expected size. Note that no cross-reaction was found with the IgGs. **(D)** Schematic representation of synaptosome enrichment protocol. **(E)** Adult mouse cortical synaptosomal protein extracts were immunoprecipitated (IP) using anti-Dyrk1a antibody and anti-IgG antibody as a negative control. The input and precipitated fractions were then resolved by sodium dodecyl sulphate–polyacrylamide gel electrophoresis (SDS–PAGE) and analyzed by Western blot using anti-Dyrk1a and anti-Dscam antibody. Note the band of 85 kD expected for the Dyrk1a protein and the 250-kD band expected for Dscam protein. No cross-reaction was found with the IgGs.

same gene were clustered into families. The sequence common to these fragments defines the selected interacting domain (Formstecher et al, 2005).

DYRK1A interaction occurs in the same ~90-AA domain of the cell-adhesion molecules encoded by *DSCAM* and its paralog *DSCAML1* (Fig 8A). DSCAM (UnitProtKB-O60469) is a 2,012 AA protein with an extracellular domain (positions 18–1,595), a transmembrane domain (1,596–1,616) and a cytoplasmic domain (1,617–2,012). The DSCAM domain interacting with DYRK1A was identified as lying between positions 1,761 and 1,850 AA (90 AA). DSCAML1 (UniProtKB - Q8TD84) is a 2,053 AA protein that also has an extracellular domain (positions 19–1,591), a transmembrane domain (1,592–1,612) and a cytoplasmic domain (1,613–2,053). The domain of DSCAM interacting with DYRK1A was identified as lying between positions 1,823 and 1,907 (84 AA). We then analyzed the interaction of the *Drosophila* homolog, minibrain (MNB), with DSCAM4. DSCAM4 (UniProtKB-B7Z0D9) (1,874 AA) has a transmembrane segment from AA 1,626 to AA 1,647, with an intracellular domain from AA 1,648 to AA 1,874. The selected interacting domain extends from AA 1,697 to AA 1,841. The location of the DYRK1A-binding domain on DSCAM suggests a phylogenetically conservation between *Drosophila* and humans (Fig 8B).

The only antibodies available against DYRK1A or DSCAM and able to give immunocytochemical signals are both from the same species (rabbit polyclonal antibodies). This precludes their use for PLA. Therefore, we confirmed the human DSCAM–DYRK1A interaction by immunoprecipitation, using adult mouse cortex (Fig 8C). We performed immunoprecipitation on a synaptosome preparation for detection of the DSCAM–DYRK1A interaction in this subcellular fraction (Fig 8D and E).

We found that DSCAM interacted with DLG1, DLG2, and DLG4, which are bona fide dendritic spine components, in both Y2H approaches and PLA (Fig 6G and H). We also found that DYRK1A interacted with STX1A, a bona fide presynaptic protein (Fig S3).

Overall, our results suggest that DSCAM–DYRK1A interactions may occur in both presynaptic and postsynaptic positions.

## Discussion

Despite the availability of various mouse models of DS, no relationship has yet been established between the cognitive impairment phenotypes found in DS and specific alterations of molecular

pathways. Furthermore, to the best of our knowledge, no specific pathways linked to synaptic alterations have been described in models overexpressing a given chr21 gene relative to models overexpressing a syntenic region.

In this study, we analyzed molecular changes in the hippocampus of two mouse models of DS. We found molecular changes linked to chromatin remodeling in the Dyrk1A BAC 189N3 mouse. By contrast, the expression of an extra copy of the entire HSA21 syntenic region, spanning ~23.3 Mb and containing 115 HSA21 coding gene orthologs (Li et al, 2007; Aziz et al, 2018), including *Dyrk1a*, on Mmu16 in the Dp(16)1Yey transgenic mouse model induced changes in glutamatergic and gaba-ergic synaptic transmission.

In large-scale Y2H experiments, both direct interactors ($n$ = 1,687) and the second-order interactors captured with our rebound screen ($n$ = 1,949) were found to be enriched in ID genes. This observation suggests that protein–protein complexes including a protein encoded by a gene on HSA21 are associated with a risk of ID.

The PLA approach can localize PPIs to a given subcellular comportment, making it possible to focus on the synaptic compartment, in which subtle deregulations may occur (Grant, 2018; Koopmans et al, 2019).

We found that interactions in the dendrite were enriched in HSA21 gene products. In particular, we were able to demonstrate that both DYRK1A and DSCAM can be present in the dendrite. DYRK1A and DSCAM are high-risk genes for ASDs, associated with an ~20-fold increase in ASD risk (Sanders et al, 2015; Willsey et al, 2018; Satterstrom et al, 2020). Changes in gene dosage for both DYRK1A and DSCAM may deregulate molecular interactions identified in network composed of 33 proteins, located in the dendritic spine, and illustrated in Fig 7.

From phalloidin staining of dendrites and dendritic spines, it is often difficult to detect PLA spots in dendritic spines. Precise location of interactions in sub-regions of dendrites will require studies using super-resolution microscopy.

Altogether, further studies aiming to decipher the molecular changes in these pathways would shed light on the pathophysiology of both DS and ASDs.

We also found a significant enrichment in ARC-dependent synaptic network proteins implicated in intelligence and brain diseases (Fernández et al, 2017). Mutations disrupting this molecular network may modify the architecture of synaptome maps, potentially accounting for the behavioral phenotypes associated with neurological and psychiatric disorders (Grant, 2018). Further studies are required to analyze the changes in protein complexes at the synapse resulting from changes in gene dosage for the various partners involved in these complexes.

DS, caused by trisomy of chromosome 21, is known to be the single commonest risk factor for early-onset AD. APP triplication has been suggested as a candidate mechanism for this phenotype, but human chromosome 21 trisomy enhances amyloid-$\beta$ deposition independently of the presence or absence of an extra copy of APP, indicating that the triplication of chromosome 21 genes other than APP probably plays an important role in AD pathogenesis in individuals with DS (Wiseman et al, 2015). We report here an enrichment in LOAD genes identified by GWAS strategies, such as RIN3, BIN1, and CASS4, in our postsynaptic network. We (Daudin et al, 2018 *Preprint*) and others (Schürmann et al, 2020) have characterized

BIN1 at the synapse by super-resolution microscopy. Other protein networks including BIN1 and CASS4 have been reported in microglia (Nott et al, 2019). These results suggest that cell-specific protein complexes may contribute to the Alzheimer phenotype in DS.

Interestingly, our Y2H and PLA approaches identified novel candidates in the postsynaptic domain, which we characterized in four layers, as described by Li et al (2016, 2017), and which has a very restricted width, in the range of 75 nm (Tao et al, 2018). Super-resolution microscopy approaches have recently revealed that spine synapses in vitro and brain slice nanodomains form a trans-synaptic column and contain discrete, precisely aligned sub-diffraction nanomodules, the number of which, rather than size, scales with spine volume (Tang et al, 2016; Hruska et al, 2018).

Changes in the stoichiometry of interactors, as expected for HSA21 proteins, may modify the functional impact of a given protein complex. The report that the same neuroligin4 mutation can generate either ID or high-level ASD supports such subtle changes (Laumonnier et al, 2004). Similarly, some protein complexes may integrate only a given form of a protein, as has been reported for TIAM1 in glutamatergic synapses from the entorhinal cortex (Rao et al, 2019).

Together, our results suggest that the PPIs identified here may occur in different dendritic spine signalosomes deregulated by the presence of three doses of HSA21 proteins.

In conclusion, we report here, for the first time, the differential impacts of chromosome 21 *DYRK1A* on chromatin remodeling and of the 115 HSA21 gene orthologs, including *DYRK1A*, on synapse function. Our results shed light on the links between DS and other forms of ID and degenerative diseases with a complex genetic basis, such as LOAD. The molecular pathways studied here could be targeted in the development of new treatments for treating the cognitive impairments of individuals with DS.

# Materials and Methods

### Animals and genotyping

All experiments were approved by the Institut National de la Santé et de la Recherche Médicale (INSERM) animal care 03882.02 and B751403 agreements (to M Simonneau), in agreement with the European community council directive 2010/63/UE.

We used wild-type mice of the OF1 strain for neuronal primary culture, wild-type of the C57BL6 strain and Tg(Dyrk1a)189N3Yah (named 189N3) or Dp(16Lipi-Zfp295)1Yey (named Dp(16)1Yey) transgenic lines for neuronal primary cultures. Genotypes were determined using genomic DNA extracted from skeletal muscle fragments.

### RNA sequencing

#### *Sample preparation*

The hippocampi were dissected from genotyped E16-E18 embryos (n = 3 or 4 per genotype for 189N3 or Dp(16)1Yey transgenic mouse, respectively). Samples were homogenized in Trizol reagent (GIBCO), purified on nucleospin column (Macherey Nagel), treated with DNase I (Ambion), and processed according to the manufacturer's instructions.

### Total stranded RNA-seq

Total-Stranded RNAseq sequencing was performed by the Centre National de Recherche en Génomique Humaine (CNRGH), Institut de Biologie François Jacob. After complete RNA quality control on each sample (quantification in duplicate on a NanoDrop 8000 spectrophotometer and RNA6000 Nano LabChip analysis on Bioanalyzer from Agilent), libraries have been prepared using the "TruSeq Stranded Total RNA" Kit from Illumina. An input of 1 µg total RNA was used for all samples, and libraries were prepared according to manufacturer's instructions. After library quality control and quantification, sample libraries have been pooled before sequencing to reach the expected sequencing depth. Sequencing has been performed on an Illumina HiSeq200 as paired-end 100 bp reads, using Illumina sequencing reagents. Fastq files produced after RNA-seq have been be processed by in-house CNRGH tools to assess quality of raw and genomic-aligned nucleotides.

### Analysis

The Bowtie-TopHat-Cufflinks pipeline was used as previously described (Trapnell et al, 2012). Reads were mapped on *M. musculus* mm10 genome and the University of California, Santa Cruz (UCSC) known genes was used as transcriptome index. Cuffmerges were run on all samples. The merged assembly were mapped on the Gencode (release M4) main annotation and all the transcripts which were not described within were removed (antisense genes, unknown transcript) to focus on protein-coding genes. For the quantification (cuffquant) of the abundance, the frag-bias-correct and the multi-read-correct options of the program on the merged assembly were used. The differential analysis was performed on two levels: gene level and transcript isoform level.

### Constructs

Mouse Dyrk1a cDNA was cloned in GFP plasmid as described (Lepagnol-Bestel et al, 2009). Human USP25 and SYK were cloned in GFP and MYC plasmids, respectively, as described (Cholay et al, 2010). Human GDI1 and DSCR9 were amplified by PCR from IMAGE: 4156714 and IMAGE: 6065320 cDNA clones, respectively (Source-Biosciences), with the following primers:

GDI1 forward: 5'-gatcggccggacgggccGACGAGGAATACGATGATCGTG
GDI1 reverse: 3'-gatcggccccagtggccTCACTGCTCAGCTTCTCCAAAGACGTC
DSCR9 forward: 5'-gatcggccggacgggccATGGGCAGGATTTGCCCCGTGAAC
DSCR9 reverse: 3'-gatcggccccagtggccTCACCATAATTCCTGTGTGTCTGAATCTGAA

The SfI digestion products of the amplicons were inserted into the multiple cloning site of the HA and GFP expression vectors respectively under control of the CMV promoter.

### Primary neuron cell cultures and transfection

Primary cultures from OF1 mice were performed as described in Loe-Mie et al (2010). Heterozygous 189N3 or Dp(16)1Yey mice were crossed with C57BL6, resulting in embryos of transgenic or wild-type genotypes. E15.5 189N3 or Dp(16)1Yey cortical neurons were dissociated by individually dissecting each embryo out of its amniotic sac, removing the head and dissecting out the target brain tissue in an separate dish. The remainder of the brain was used for genotyping. Neurons from each embryo were dissociated enzymatically (0.25% trypsin), mechanically triturated with a flamed Pasteur pipette, and individually plated on 24-well dishes (1 × 105 cells per well) coated with poly-DL-ornithine (Sigma-Aldrich), in DMEM (Invitrogen) supplemented with 10% fetal bovine serum. 4 h after plating, DMEM was replaced by Neurobasal medium (Invitrogen) supplemented with 2 mM glutamine and 2% B27 (Invitrogen). For nuclear interactions or dendritic interactions, cortical neurons were analyzed after 7 or 21 d in culture, respectively.

Cortical or hippocampal primary neurons were cultured as described above. At DIC5, the cells were transfected with constructs using Lipofectamine 2000 (Invitrogen), as described by the manufacturer. Cells were analyzed 48 h after transfection at DIC7.

### HEK293 cell cultures and transfection

HEK293 cell line were plated in 24-well plates in DMEM (Invitrogen) supplemented with 10% fetal bovine serum. At 70% confluency, the cells were transfected with constructs (co-transfections were performed at 1:3 ratio) using Lipofectamine 2000 (Invitrogen), as described by the manufacturer. Cells were analyzed 48 h after transfection.

### In situ PLAs and microscopy

Cells were fixed by incubation for 20 min at room temperature in 4% paraformaldehyde in PBS, permeabilized by incubation for 10 min at room temperature in 0.3% Triton X-100 in PBS, washed two times within PBS, and PLA was realized according to the instructions of the manufacturer (DuoLink, Sigma-Aldrich). Primary antibodies used were as shown in Table S4. For the analysis of PLA interactions points, cells were scanned using the laser scanning confocal microscope (Leica, SP5 from PICPEN imagery platform Centre de Psychiatrie et Neuroscience) at 63× magnification, and Z-stacks were build using the ImageJ software (Wayne Rasband, NIH). Nuclear PLA interaction number was manually counted inside the heterochromatin and normalized with the nuclear area of each neuron. Synaptic PLA interaction number was manually counted on 150-µm-long dendritic segments starting after the first branch point in the dendritic tree.

### Statistical analysis

The analyses performed on transgenic neurons with at least three embryos and at least 10 cells per embryo for synaptic and nuclear analyses. The analyses performed on OF1 neurons with at least three different cultures and at 8 cells and 14 cells per culture for synaptic and nuclear analyses, respectively. The analyses performed on HEK293 cells with at least three different transfections and 25 cells per transfection for nuclear or cytoplasmic analyses.

Statistics were performed using IgorPro (Wavemetrics) and Excel Software. Normality was checked by visual examination of data graphic representations. Results are reported as mean ± SEM. Comparisons between two groups were performed using unpaired

two-tailed Student's t tests (*$P$ < 0.05, **$P$ < 0.01, ***$P$ < 0.001, ****$P$ < 0.0001).

## Protein extraction and Western blot analysis

HEK293 cells or mouse cortex (pool from three adult OF1 mice) were homogenized on ice in Tris-buffered saline (100 mM NaCl, 20 mM Tris–HCl, pH 7.4, 1% NP40, 1× CIP). The homogenates were centrifuged at 13,000$g$ for 10 min at 4°C and the supernatants were stored at −80°C. Cell lysate protein concentration was determined using the BCA Protein assay kit (Thermo Fisher Scientific). For SDS–PAGE, 40 $\mu$g of protein was diluted in Laemmli 1× (Bio-Rad) with DTT and incubate for 5 mn at 95°C. Proteic samples were loaded in each lane of a 4–15% precast polyacrylamide gel (Bio-Rad) and ran in Mini-Protean at 200V in Tris/Glycine running buffer (Bio-Rad). After SDS–PAGE, proteins were semi-dry electroblotted onto nitrocellulose membranes using the Trans-Blot Turbo Transfer System (Bio-Rad). Membranes were incubated for 1 h at room temperature in blocking solution (PBS 1× containing 5% non-fat dried milk, 0.05% Tween 20) and then for overnight at 4°C with the primary antibody. Primary antibodies used were as shown in Table S4. Membranes were washed in PBS 1× containing 0.05% tween 20 and incubated for 1 h at room temperature with anti-mouse, anti-rabbit or anti-goat HRP-conjugated secondary antibody. Membranes were washed three times in PBS 1× containing 0.05% tween 20. Immune complexes were visualized using the Clarity Western ECL Substrate (Bio-Rad). Chemiluminescence was detected using the ChemiDoc XRS Imaging System (Bio-Rad). As secondary antibodies, we used protein A or protein G IgG, HRP-conjugated whole antibody (1/5,000; Abcam ab7460 or ab7456, respectively).

## Immunoprecipitation

1 mg of protein extracts were incubated, after preclear with 50 $\mu$l of dynabeads (Novex), 3 h at 4°C under rotating with 10 $\mu$g of primary antibody (Table S4; anti-mouse and anti-rabbit whole IgG [Millipore 12-371 and 12-370, respectively]). Add 50 $\mu$l of protein A or protein G dynabeads and incubate 30 mn at 4°C under rotating. Protein–antibody complexes were washed four times in 100 mM NaCl, 20 mM Tris–HCl, pH 7.4, 1% NP40, and analyzed by immunoblot.

## Laser-assisted microdissection, total RNA preparation, and quantitative real-time PCR (Q-RT-PCR) analysis

Embryonic left and right hippocampus was microdissected from genotyped P21 mouse brains using a laser-assisted capture microscope (Leica ASLMD instrument) with Leica polyethylene naphthalate membrane slides as described in Lepagnol-Bestel et al (2009). RNA preparation and Q-RT-PCR are performed as described in Lepagnol-Bestel et al (2009). Q-RT-PCR results are expressed in arbitrary unit.

Reagents Stock solutions were prepared in water or DMSO, depending on the manufacturers' recommendation, and stored at −20°C. Upon experimentation, reagents were bath applied following dilution into artificial cerebrospinal fluid (ACSF) (1/1,000). ACSF was purchased from Sigma-Aldrich.

## Preparation of synaptosomes and protein extraction

Cortex from 3 to 4 mo mice brains were dissected and homogenized (pool of six animals) in H buffer (0.32M sucrose, 5 mM Hepes 1M, pH 7.4, and 1 mM EDTA) using a glass potter. The homogenate was centrifuged at 800$g$ for 7 mn to remove nuclei and debris, the supernatant was centrifuged at 9,200$g$ for 10 mn to remove cytosolic supernatant. The pellet was resuspended in H buffer and gently stratified on a discontinuous Percoll gradient (5%, 10% and 23% vol/vol in H-buffered Percoll) and centrifuged at 20,000$g$ for 11 mn. The layer between 10% and 23% Percoll (synaptosomal fraction) was collected and washed in H buffer by centrifugation. The synaptosomal pellets were resuspended in MLB buffer (1% NP40, 100 mM NaCl, and 20 mM Tris, pH 7.4, in PBS with 1× protease and phosphatase inhibitor cocktail) for 10 min on ice and centrifuged 15 min at 10,000$g$ at 4°C. The supernatants were stored at −80°C until used and lysate protein concentration was determined using the DCTM Protein assay (Bio-Rad).

## Network bioinformatics analyses

Amigo2 was used as a tool for searching and browsing the GO database (http://amigo.geneontology.org/amigo).

We used Disease Association Protein–Protein Link Evaluator (DAPPLE) that looks for significant physical connectivity among proteins encoded for by genes in loci associated to disease (Rossin et al, 2011). Interactions are extracted from the database "InWeb" that high confidence interactions. Connections can be direct and indirect. The significance of the interaction parameters are tested using a permutation method that compares the original network with thousands of networks created by randomly re-assigning the protein names while keeping the overall structure (size and number of interactions) of the original network.

To complement the DAPPLE analysis, we used the WebGestalt suite (Liao et al, 2019), String: functional protein association networks (string-db.org) and Syngo: Synaptic Gene Ontologies and annotations consortium—An evidence-based, expert-curated resource for synapse function and gene enrichment studies (Koopmans et al, 2019).

The analysis of contingency tables was performed using a Fisher's exact test.

## Yeast two-hybrid experiments

A list of 234 genes from Hsa21 was examined.

### Y2H library

We used a human Adult brain poly(A+) RNA (Invitrogen: Discovery Line Human normal Brain mRNA, Sex: M, Age: 27, Cat. no.: D6030-15, LOT No: A308079) constructed in the pP6 plasmid derived from the original pACT222 and transformed in *Escherichia coli* (DH10B; Invitrogen). The complexity of the primary libraries was over 50 million clones. Sequence analysis was performed on 300 randomly chosen clones to establish the general characteristics of each library. The libraries were then transformed into yeast by classical lithium acetate protocol. Ten million independent yeast colonies

were collected, pooled and stored at –80°C as equivalent aliquot fractions of the same library (Fromont-Racine et al, 1997, 2002).

Two-hybrid screens were performed using a cell to cell mating protocol. For each bait, a test screen was performed to adapt the screening condition. The selectivity of the HIS3 reporter gene was eventually modulated with 3-aminotriazole (Sigma-Aldrich) to obtain a maximum of 285 histidine-positive clones for 50 million diploids screened. For all the selected clones, lacZ activity was estimated by overlay assay on solid media in 96-well plate format. Inserts of all positive clones were amplified by PCR22, 23 and then sequenced on an ABI 3700 automatic sequencer (Applied Biosystem).

### Prey identification

5′ and 3′ sequences were determined for all positive clones in a screen. These were in turn filtered for quality using PHRED and ALU repeats were masked. Sequence contigs were built using CAP324 and searched against the latest release of GenBank using BLASTN.

### Identifying reliable interactions

Interactions were filtered based on a PBS (Fromont-Racine et al, 2002) The PBS was calculated based on randomly sequenced cDNA library and adopts the conventional form of a *P*-value, where the smaller the PBS (*P*-value) the more significant. The PBS relies on two different levels of analysis. First, a local score takes into account the redundancy and independency of prey fragments, as well as the distribution of reading frames and stop codons in overlapping fragments. Second, a global score takes into account the interactions found in all the screens performed at Hybrigenics using the same library. This global score represents the probability of an interaction being nonspecific. For practical use, the scores were divided into four categories, from A (highest confidence) to D (lowest confidence). A fifth category (E) specifically flags interactions involving highly connected prey domains previously found several times in screens performed on libraries derived from the same organism. Finally, several of these highly connected domains have been confirmed as false positives of the technique and are now tagged as F. The PBS scores have been shown to positively correlate with the biological significance of interactions.

### Preparation of bait constructs

The coding sequence for each bait protein was PCR-amplified and cloned in-frame with the LexA DBD into plasmid pB27, derived from the original pBTM116. DBD constructs were checked by sequencing the entire insert. Several inserts were cloned in-frame with the Gal4 DBD into plasmid pB66, derived from combines data from a variety of public PPI sources including MINT, BIND, IntAct, and KEGG and defines pAS2ΔΔ (24). For DSCR8 (contested coding gene) cDNA coding for MKEPGPNFVTVRKGLHSFKMAFVKHLLLFLSPRLECSGSITDHCSLHLPV-QEILMSQPPEQLGLQTNLGNQESSGMMKLFMPRPKVLAQYESIQFMP have been used.

### Preparation of Dyrk1a ΔpolyHis mutant bait construct

The sequence coding for Dyrk1a C terminus (aa 600–763) was modified to remove the poly-histidine stretch to prevent potential artefacts of binding with Cysteine-rich prey proteins without altering the folding of the bait. This region was modify by gene

synthesis (Eurofins-Genomics) inserted in the cDNA cloned resulting in the sequence behind: DYRK1A delta polyHis: PQQNALHAAHGNSSAAAGAHAGAAHAHGQQALGNRTRP.

Preparation of prey constructs 1-by1 assays DLG1 (aa 305–653), DLG4 (154–356), DLG2-5 (aa 84–440) prey plasmids were extracted from the diploid cells obtained from the Y2H screening with wild-type DSCAM of the Human Adult Brain library. Inserts are cloned in-frame with the Gal4 activation domain (AD) into plasmid pP6, derived from the original pGADGH. The coding sequence for DLG3 (aa 212–385) was PCR-amplified and cloned in-frame with the Gal4 AD into plasmid pP7. The AD constructs were checked by sequencing.

### Y2H screening and 1by1 interaction assays

Bait and prey constructs were transformed in the yeast haploid cells, respectively, CG1945 or L40ΔGal4 (mata) and YHGX13 (Y187 ade2-101::loxP-kanMX-loxP, matα) strains. The diploid yeast cells were obtained using a mating protocol with both yeast strains (Fromont-Racine et al, 2002) His+ colonies were selected on a medium lacking tryptophan, leucine, and histidine, and supplemented with 3-aminotriazole to handle bait autoactivation when necessary. The prey fragments of the positive clones were amplified by PCR and sequenced at their 5′ and 3′ junctions. The resulting sequences were used to identify the corresponding interacting proteins in the GenBank database (NCBI) using a fully automated procedure.

Interaction pairs were tested in duplicate as two independent clones from each diploid were picked for the growth assay. For each interaction, several dilutions (10-1, 10-2, 10-3, and 10-4) of the diploid yeast cells (culture normalized at $5 \times 10^4$ cells) and expressing both bait and prey constructs were spotted on several selective media. The DO-2 selective medium lacking tryptophan and leucine was used as a growth control and to verify the presence of both the bait and prey plasmids. The different dilutions were also spotted on a selective medium without tryptophan, leucine, and histidine (DO-3). Four different concentrations of 3-AT, an inhibitor of the HIS3 gene product, were added to the DO-3 plates to increase stringency. The following 3-AT concentrations were tested: 1, 5, 10, and 50 mM.

## Data Availability

DEG data from embryonic hippocampus of 189N3 and Dp(16)1Yey/+ are available at GEO under the accession number GSE201290. Y2H data are deposited in Zenodo (10.5281/zenodo.6902878; Viard et al., 2022), and will be available at Intact, EBI-EMBL under the accession number IM-27626.

## Supplementary Information

## Acknowledgements

We would like to acknowledge the funding support from the European Union's Seventh Framework Programme for research, technological development, and demonstration AgedBrainSYSBIO under grant agreement

No. 305299 (to M Simonneau) and Fondation Lejeune. This work was also partly funded by INSERM, European JPND (TransPathND; ANR-17-JPCD-0002), and CNES (to M Simonneau).

## Author Contributions

J Viard: formal analysis and experimental work.
Y Loe-Mie: formal analysis.
R Daudin: formal analysis and investigation.
M Khelfaoui: formal analysis and investigation.
C Plancon: formal analysis.
A Boland: formal analysis.
F Tejedor: resources.
RL Huganir: resources.
E Kim: resources.
M Kinoshita: resources.
G Liu: resources.
V Haucke: resources.
T Moncion: software and formal analysis.
E Yu: resources.
V Hindie: software, formal analysis, and investigation.
H Bléhaut: resources.
C Mircher: resources.
Y Herault: resources.
J-F Deleuze: software.
J-C Rain: resources.
M Simonneau: conceptualization and writing—original draft, review, and editing.
A-M Lepagnol-Bestel: conceptualization, formal analysis, and writing—original draft, review, and editing.

## Conflict of Interest Statement

The authors declare that they have no conflict of interest.

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
