## [Reviewer comments · Life Science Alliance]

Life Science Alliance

Chr21 protein-protein interactions: enrichment in products involved in ID, autism and LOAD

Julia Viard, Yann Loe-Mie, Rachel Daudin, Malik Khelfaoui, Christine Plancon, Anne Boland, Francisco Tejedor, Richard Haganir, Eunjoon Kim, Makoto Kinoshita, Guofa Liu, Volker Haucke, Thomas Moncion, Yuejin Yu, Valerie Hindie, Henri Blehaut, Clotilde Mircher, Yann Herault, Jean-François Deleuze, Jean Rain, Michel Simonneau, and Aude-Marie Lepagnol-Bestel

DOI: <https://doi.org/10.26508/lsa.202101205>

Corresponding author(s): Michel Simonneau, ENS Paris-Saclay and Aude-Marie Lepagnol-Bestel, INSERM

Review Timeline:	Submission Date:	2021-08-20
	Editorial Decision:	2021-10-04
	Revision Received:	2022-05-03
	Editorial Decision:	2022-06-07
	Revision Received:	2022-07-04
	Accepted:	2022-07-06

Scientific Editor: Novella Guidi

Transaction Report:

October 4, 2021

Re: Life Science Alliance manuscript #LSA-2021-01205-T

Michel Simonneau
Ecole Normale Supérieure Paris-Saclay
61 Avenue Président Wilson
Cachan 94230
FRANCE

Dear Dr. Simonneau,

Thank you for submitting your manuscript entitled "Chr21 protein-protein interactions: enrichment in products involved in ID, autism and LOAD" to Life Science Alliance. The manuscript was assessed by expert reviewers, whose comments are appended to this letter. As you will note from the reviewers' comments below, the reviewers do raise some concerns. Reviewer 1 list numerous critiques to be addressed, including over-interpretation of the results thus suggesting substantially rewriting the manuscript with the focus on the interaction networks. Reviewer 3 requests the authors to pare down the text to make it easier to read and ask to provide more information to validate the PLA assays. We, thus, encourage you to submit a revised version of the manuscript back to LSA that responds to all of the reviewers' points.

We invite you to submit a revised manuscript addressing the Reviewer comments.

Thank you for this interesting contribution to Life Science Alliance. We are looking forward to receiving your revised manuscript.

Sincerely,

B. MANUSCRIPT ORGANIZATION AND FORMATTING:

Reviewer #1 (Comments to the Authors (Required)):

In the work by Viard and colleagues, the authors offer an interesting and valuable resource tool by analyzing in-depth binary protein interactions of human chromosome 21 coding genes using large-scale yeast-two-hybrid screens. Some of the interactions are further validated by PLA with an emphasis on interactions occurring at the synapse. The interaction networks are discussed in the context of the particular enrichment of proteins encoded by genes associated with intellectual disabilities and with synaptic alterations related to Down syndrome neuropathological traits. However, there are several aspects that weaken the manuscript. These include missing data (e.g., definition of the interaction domain in DSCAMs with DYRK1A, interaction between the Drosophila DSCAM and DYRK1A orthologs, quantification of the PLA data), lack of rigor in the use of scientific language, incorrect labeling/missing/repetition of Figures, over-interpretation of the results, and a difficult-to-read manuscript full of mistakes (the use of abbreviations is very messy). In addition, I find the results associated to Figure 1 unconnected from the rest of the manuscript. They are not needed for understanding or discussing the rest of the data and this part of the work falls short of providing any further validation or analysis even at a descriptive level: e.g., changes in differential gene expression during development and in the adult, comparison between transcriptome and proteome.

I suggest substantially rewriting the manuscript with the focus on the interaction networks and I ask the authors to make extra efforts on the written and graphical presentation of the results.

Abstract: The summary of the project in the Abstract, as written, is difficult to understand and it is full of inaccuracies and grammatical errors, some of which are listed below:

- The first sentence of the Abstract is not right since there are partial chromosome 21 trisomies in Down syndrome individuals. Moreover, it is wrong to state "234 genes on Chr21": there are currently 660 genes mapped to the human chromosome (Ensembl release 104 - May 2021).
- It is not right to use "exome sequencing" for describing RNA-Seq analysis. The term is also used in the Introduction section.
- Not clear what the 154 distinct YTH screens are using as baits
- The sentence "Hsa21-encoded proteins are located at the dendritic spine postsynaptic density in a protein network located at the dendritic spine post synapse" implies that all Hsa21-encoded proteins are located at dendritic spines.
- Chr21 and Hsa21 are both used as abbreviation for human chromosome 21.
- Not clear the meaning of the sentence "Hsa21 DYRK1A and DSCAM that confers a ~ 20-fold increase in Autism Spectrum Disorders (ASDs)".
- The authors write "a DSCAM intracellular domain binds either DYRK1A or DLGs". No experimental data is shown to support this statement.
- The sentence "The DYRK1A-DSCAM interaction is conserved from drosophila to humans." would imply a broad analysis of the interaction between the two proteins in many different organisms. Only proteins from Drosophila and humans are analyzed. In fact, the results for the interaction between the Drosophila proteins are not shown at all.

Materials and Methods. The authors should carefully check the section. Some issues are listed below, and I have also included some others in other parts of the Review.

- The subsection "Primary cell cultures and transfection" should be merged with the subsection "Primary cell cultures".
- In the case of HEK293 cells, 5 cells per transfection are very low numbers for the analysis performed, which is not clear when it has been used along the manuscript.
- The two subsections named "Statistical analysis" should be merged, and the authors check which one provides the right information. A test to check the normality of the data should be applied before using a parametric test or a non-parametric test.
- Some abbreviations are not properly annotated: ACSF, DCGIV, D-APV.
- What is a "cutting solution"?
- The references cited in the Materials & Methods section are listed outside the Reference sections, is this OK? Anyway, some of them are missing: Cholay et al. 2010.

Introduction.

- A reference for the incidence of DS should be provided: world-wide?, developed countries?
- DS-associated trisomy 21 could affect the whole chromosome or be partial. Please amend accordingly.
- Which is the source of the 234 genes present in human chromosome 21?. ENSEMBL lists currently a total of 660 genes: 236 coding genes and 424 non coding genes. Why the authors are only referring to the coding genes as responsible for the DS-associated phenotypes? There are no reasons for such assumption.
- Not clear to me which are the reasons underlying the choice of only 3 references (Ahn, Thomazeau, Smith) among the many available to sustain the involvement of the DYRK1A gene as a contributor to some of the neuropathological traits of DS.

Results, Subsection 1: As mentioned above, the data of this subsection is unnecessary in the context of the rest of the manuscript and it does not reach the level of quality of the other parts of the manuscript.

- The title mention "quantitative proteomics" but no data is included.
- It will be very informative to provide an updated list of the mouse genes included in the Dp(16)1Yey transgenic mice and their corresponding human orthologs, so the readers have clear information on which are the 119 genes triplicated. This will help the reader to navigate among the different gene contexts of the many DS mouse models. The reference for this transgenic model is Li et al, 2017, PMID:17412756, and not Yu et al., 2010.
- Information on the preparation of the libraries for RNA-Seq as well as for the type of sequencing performed (single-end?, paired-end?, length of reads?) should be provided. Which mice are used as controls in the RNA-Seq experiments? euploid littermates? Do the pools contained male and female animals? or are gender specific?
- Information on the statistical analysis used for the differential analysis of the RNA-Seq data and the parameters applied for the cut-off should be provided.
- There is no proper comparison of the two set of DEG genes. They are supplied in the form of Supplementary data as pdf not easy to digest by the readers. Supp Table S1 and S2 are misassigned in the text to each of the mouse models
- The RNA-Seq experiments data are not deposited in publicly available sites.
- The RNA-Seq does not detect changes in the mouse chromosome 16 triplicated genes in the Dp(16)1Yey model. A published analysis of E16 Dp(16)1Yey forebrains showed that most of the few deregulated genes with increased expression mapped to the triplicated region (Guedj et al., 2018, PMID: 27586445; Aziz et al., 2018, PMID: 29716957).
- The reference Loe-Mie et al., 2010, does not apply to the statement that "Deregulation of chromatin proteins is in agreement with published data from the authors linking changes in DYRK1A expression and alteration of chromatin-regulatory proteins.
- The sentence "Deregulation of chromatin proteins..." should say "Deregulation of the expression of genes encoding for chromatin proteins...". This also applies to the next sentence for Dp(16)1Yey/+.
- References Kleschevnikov et al., 2012; Raveau et al., 2018 refer to mouse models of DS not to DS individuals.
- The quality of Figure 1 is low, there is no explanation for the different colors, if any, and the lettering is too small to properly convey any information. I do not agree with the authors in suggesting a network of protein-protein interaction changes based on deregulation of mRNA. Changes in transcripts do not necessarily translate into changes in protein amounts. Second, more than a connection among the proteins encoded by the mRNAs, the DEGs should illustrate altered signaling pathways with specific transcription factors that use these genes as transcriptional targets. An example of such regulation can be found in the analysis of the adult hippocampus of the DS Tc1 mouse model (Granno et al., 2019; PMID: 31086297). In my opinion, the authors have chosen not the best representation of their findings for differential GO term enrichments in the two mouse models analyzed. The Programme and the parameters used for the GO analysis should be indicated in the Materials and Methods section.
- Figure 2 is never mentioned in the text.
- The authors mention in the Discussion the uniqueness of the approach in comparing differential gene expression in DS mouse models. However, at least two published articles have already done that (Guedj et al., 2018, PMID: 27586445; Aziz et al., 2018, PMID: 29716957).

Results, Subsection 2:

- A list of the baits for the primary (chromosome 21 baits) and secondary (rebounds baits) Y2H screens should be included with an entry number of the cDNA (NCBI or Ensembl ID) used for each of the genes to provide information on the specific isoform cloned for each gene. Likewise, the reader will greatly benefit for having the results of the Y2H interactions in a workable format (for instance, Excel with paired interactions). The information of the two tables will be a valuable source for many researchers.
- A clear description of the categories a-f shown in Figure 3 should be provided in the Figure Legend. There are mistakes in the assignment of the letters of the individual panels, please check. The authors should make clear in the text that most of the novel interactions detected belong to the "lowest confidence" category, while the trend reversed when the already described baits are analyzed. Indeed, there is not experimental validation by orthogonal approaches providing some numbers for the expected percentage of non-specific interactions in this category. Therefore, a note of caution should be included.
- The Figure legend has to include definitions for S2, S10 and S11 used in panels E and F. A definition in the text will be also welcomed. In fact, Supplementary Table S3, including the lists of the genes, has a format non-informative at all. What does X21 refer to? Which is the statistical test used for calculating the p-value for the enrichment in Figure 3E and F?
- As a piece of valuable information, the authors should include the list of genes that are responsible for the enrichment in categories shown in Figure 4, as a supplementary Excel Table. The Programme used for the analysis is missing in the Materials and Methods section.
- The authors might mean "We performed" instead of "We realized" (page 11, last paragraph).
- Does Figure 4 include only the chromosome 21 baits? If so, it should be indicated.

- Within the Y2H experiments subsection in the Materials and Methods section a determination of Orthologs collection is mentioned. With what intention? how were they determined?.

Results, Subsection 3: Although the information included in this subsection represents the validation of some of the detected YTH interactions by an independent approach, it acts a little bit as a disruptor for the main story line, which focuses on events occurring at the synapse. The two examples included in the subsection relate to putative nuclear interactions.

- The authors could add the better known name of DROSHA for RNASEN (this would make understandable the sentence "The complex is minimally composed of the ribonuclease enzyme Drosha"), and Clusterin for CLU.
- In Figure 5A, the authors could include arrows to indicate the direction of the interactions detected in the Y2H screen (double arrow in case, the interaction was confirmed in the rebound screens). Please note that TO-PRO-3 stains DNA, not nuclear bodies. The big nuclear stain dots correspond to heterochromatin, highly evident in mouse cells.
- Page 16, last paragraph, it should say For DYRK1A-RNASEN interaction.
- I do not agree with the hypothesis raised by the authors in which DGRC8 haploinsufficiency may occur by titration of its partner RNASEN: first, the co-IP shown by the authors could be interpreted either by the formation of trimolecular complexes, or by the formation of independent paired interactions (RNASEN-DGRC8, RNASEN-DYRK1A) and at this point no competition among the three proteins has been demonstrated. In fact, the relative enrichment of the proteins in the IP experiment indicates that DYRK1A participate in a small pool of the RNASEN interactions in comparison with those established with DGRC8. I would eliminate the sentence, and leave the one regarding to putative links to alterations in the microRNA network, which might be a functional consequence of any type of alterations in the biology of RNASEN as result of DYRK1A physical interaction (interaction with partners, activity, stability).
- Figure 5C and D are also included as Supplementary Figure S1A, B and C. Likewise, Figure 5A and B are also included as Supplementary Figure S3.

Results, Subsection 4: This subsection offers a nice piece of information providing validation of the interactions of different chromosome 21 encoded proteins with important players at the synapsis, highlighting two important aspects: the validation of some of the direct interactions detected by the Y2H in a physiological context and with endogenous proteins and the highlight of the synapse as an important physiological target of chromosome 21 encoded proteins.

- As in the case of Figure 5, the panel A in Figure 6 could include arrows to indicate the direction of the detected interactions (double for those cases shown in both directions). The label of y-axis in the two bar graphs included in Figure 6 is missing.
- Not all the PLA data shown in Figure 6 has been quantified and they should. In fact, some of the images do not allow to distinguish whether some interactions are positive or not (e.g., KNCJ6-DLG4, HUNK-SYNPO or ITS1-DLGAP1).
- The authors mention Supplementary Figure S1 in regard to data shown in Figure 6D, which includes no such data. Likewise, Supplementary Figure S2 does not contain data related to Figure 6G-H.
- Which is the explanation for the increase PLA signal shown in Figure 6D for KNCJ6-DLG2 interactions in the DS mouse model: increased KNCJ6 protein expression because of the triplication of the gene? increased amounts of both partners?
- Have any of the YTH interactions analyzed by PLA not been validated?

Results, Subsection 5:

- How the authors know that the interacting region of DSCAM and DSCAML1 with DYRK1A maps to the intracellular domain? (the one shown in Figure 7C, I assume). The experimental data have to be shown.
- How the authors know that the corresponding orthologs in Drosophila do interact? The experimental data has to be shown.
- The panels in Figure 7 are wrongly mentioned along the corresponding text in the manuscript. It looks like that the text refers to a different layout of the Figure.
- Is the DSCAM-DYRK1A interaction in the synapsomal fraction detected in a reverse co-IP (IP with an anti-DSCAM antibody)? Thus, the authors might prove the presence of DSCAM and DLG complexes using the co-IP method and the synaptosomes as fraction. This will provide some qualitative information on the relative presence of these proteins in association with DSCAM.
- The methods for the preparation of synaptosomes should be included in the Materials and Methods section.
- The authors stress the fact that DSCAM and DYRK1A interact in the postsynapse (Results and Discussion). However, I understand that the subcellular fraction used for the co-IP experiments corresponds to whole synaptosomes thereby including pre- and post-synapses. If the fractionation method renders mostly postsynapses, the enrichment should be experimentally proven. If not, the text needs to be modified.

Results, Subsection 6: Which are the differences between Figure 8 and Figure 6A? Figure 8 looks like just a repetition of that panel. In fact, this subsection is a kind of summary of results that does not stand as a Results paragraph in itself. It could be included as part of the subsection 4 or the Discussion.

Reviewer #2 (Comments to the Authors (Required)):

This ms by Viard et al. aims to decipher pathways associated to cognitive impairment phenotypes in two different DS mice models: Dp(16)1Yey, with an extra copy of the entire Hsa21 syntenic region, and 189N3 with an extra copy of Dyrk1A gene. Transcriptomics analysis seems to indicate different activated pathways for each genotype: chromatin related genes for the Dyrk1A model and a synapse related genes for the Dp(16)1Yey model. Y2H and further PLA analysis identified Chr21 proteins

in PSD that correlate with intellectual disabilities in DS. Since the main focus of the paper is to decipher intellectual disabilities related pathways in DS models there is a considerable amount of data that need to be included

These are the main concerns:

Figure 1

The described interactomes were constructed including all the deregulated genes. Perhaps a separated analysis of unregulated and down regulated genes could give a more informative perspective

Figure 2

Several Dp16 up regulated genes were associated to synaptic transmission and glutamate receptor pathway. There are also 65 down regulated genes: were they related to any particular pathway?

Figure 3 and 4

Due to the main focus in DS models, figures 3 and 4 should be included as supplementary data, despite the huge amount of data and effort.

Figure 5

In PLA strategy, DSCR9 was expressed as a GFP fusion protein since there are no antibodies available. That is probably due to the fact that DSCR9 is described as a regulatory lncRNA

Figure 6

It is not explained what mice genotype is the source of the samples used to generate the data B to G. Then, there is a change in the nomenclature in figure legend with no link to the figure. The general organization of the data is confusing. Described PPI need to be quantified in both DS models in order to test their relevance in ID.

Figure 7

Similar to the previous figure, DSCAM-DYRK1A interaction need to be addressed in both DS models.

Figure 8

Should either be included as a supplementary figure and commented directly in the discussion or stay at the Results section and be confronted with quantifications in both DS models.

Reviewer #3 (Comments to the Authors (Required)):

This study by Viard et al., studied two mouse models of Down syndrome using exome sequencing, large-scale yeast-two-hybrid and proximity ligation assays. The data point to the involvement of chromatin remodeling proteins and synaptic proteins in Down syndrome. These datasets can be of relevance for future studies aimed to better understand Down syndrome. Overall, the datasets seem robust and the comparison between the two mouse models is interesting, but the provided data is merely descriptive. The manuscript is quite dense and at times difficult to read. It could benefit from editing to make the flow more coherent. For instance, the results section contains considerable discussion and speculation that could be omitted and left for the discussion.

I have a number of concerns with the PLA assays that should be validated more extensively to be convincing:

- the phalloidin staining in these experiments gives little context, individual spines are difficult to distinguish. This makes it hard to conclude that these interactions are indeed taking place in spines. PLA spots seem to localize throughout the dendrites, many PLA spots even seem to localize to extracellular sites (e.g., KCNJ6-DLG2 in Figure 6D).
- the quality of the PLA experiment relies heavily on the quality of the antibodies that are used. To what extent are these antibodies validated?
- what are the negative controls in the different panels? This could be indicated more clearly.
- related to all these points, quantification of PLA spots seems critical to convey the specificity of this analysis. The bar graph in 6G might do this for a few, but since it has no label it is difficult to interpret.

Page 16: "EK293" > "HEK293"

We thank the three reviewers for their constructive comments. We reorganized the resource manuscript and the Figures accordingly: two principal Figures (Figure 1 and Figure 2) have been changed for clarity, 7 supplementary Figures were added (with now 9 Supplementary Figures), 3 Supplementary Tables were added and the main text was corrected for English.

Reviewer #1 (Comments to the Authors (Required)):

In the work by Viard and colleagues, the authors offer an interesting and valuable resource tool by analyzing in-depth binary protein interactions of human chromosome 21 coding genes using large-scale yeast-two-hybrid screens. Some of the interactions are further validated by PLA with an emphasis on interactions occurring at the synapse. The interaction networks are discussed in the context of the particular enrichment of proteins encoded by genes associated with intellectual disabilities and with synaptic alterations related to Down syndrome neuropathological traits. However, there are several aspects that weaken the manuscript. These include missing data (e.g., definition of the interaction domain in DSCAMs with DYRK1A, interaction between the Drosophila DSCAM and DYRK1A orthologs,

Interaction domains are presented for DSCAM, DSCAML1 and DSCAM4 (the Drosophila ortholog) page 18 of the manuscript and illustrated in Figure 8 A & B (pages 37-38).

quantification of the PLA data)

Quantification of PLA data is now presented in Supplementary Figure S9.

lack of rigor in the use of scientific language, incorrect labeling/missing/repetition of Figures, over-interpretation of the results, and a difficult-to-read manuscript full of mistakes (the use of abbreviations is very messy).

We carefully checked these points.

In addition, I find the results associated to Figure 1 unconnected from the rest of the manuscript. They are not needed for understanding or discussing the rest of the data

and this part of the work falls short of providing any further validation or analysis even at a descriptive level: e.g., changes in differential gene expression during development and in the adult, comparison between transcriptome and proteome.

I suggest substantially rewriting the manuscript with the focus on the interaction networks and I ask the authors to make extra efforts on the written and graphical presentation of the results.

We rewrote related parts of the manuscript accordingly, focusing on interaction networks. We replaced Figure 1 and Figure 2 by new Figures showing simpler networks based on String analysis and on SynGO analysis (Koopmans et al., 2019), respectively. SynGO analysis was recently reported to show the implication of fundamental processes related to neuronal function, including synaptic organization, differentiation and transmission (Trubeskoy et al., Nature, 2022).

Abstract: The summary of the project in the Abstract, as written, is difficult to understand and it is full of inaccuracies and grammatical errors, some of which are listed below:

The main manuscript was corrected for English.

- The first sentence of the Abstract is not right since there are partial chromosome 21 trisomies in Down syndrome individuals. Moreover, it is wrong to state "234 genes on Chr21": there are currently 660 genes mapped to the human chromosome (Ensembl release 104 - May 2021).

The point of partial HSA21 trisomies was corrected. We indicated (page 4):

This chromosome carries 235 protein-coding genes and 441 non-protein-coding genes (Ensembl release 106 – April 2022).

- It is not right to use "exome sequencing" for describing RNA-Seq analysis. The term is also used in the Introduction section.

The term RNA-Seq analysis is now used instead of exome sequencing .

In the M&M, we added a section related to RNA-seq method used (page 40):

Total stranded RNA-seq Total Stranded RNAseq sequencing was performed by the Centre National de Recherche en Génomique Humaine (CNRGH), Institut de Biologie François Jacob, Evry, FRANCE). After complete RNA quality control on each sample (quantification in duplicate on a NanoDrop™ 8000 spectrophotometer and RNA6000 Nano LabChip analysis on Bioanalyzer from Agilent), libraries have been prepared using the “TruSeq Stranded Total RNA” Kit from Illumina. An input of 1 µg total RNA was used for all samples, and libraries were prepared according to manufacturer’s instructions. After library quality control and quantification, sample libraries have been pooled before sequencing to reach the expected sequencing depth. Sequencing has been performed on an Illumina HiSeq200 as paired-end 100 bp reads, using Illumina sequencing reagents. Fastq files produced after RNA-seq sequencing have been processed by in-house CNRGH tools in order to assess quality of raw and genomic-aligned nucleotides.

- Not clear what the 154 distinct YTH screens are using as baits
- The sentence "Hsa21-encoded proteins are located at the dendritic spine postsynaptic density in a protein network located at the dendritic spine post synapse" implies that all Hsa21-encoded proteins are located at dendritic spines.

These points were corrected.

- Chr21 and Hsa21 are both used as abbreviation for human chromosome 21.

We now use HSA21 in the text.

- Not clear the meaning of the sentence "Hsa21 DYRK1A and DSCAM that confers a ~ 20-fold increase in Autism Spectrum Disorders (ASDs)".
- The authors write "a DSCAM intracellular domain binds either DYRK1A or DLGs". No experimental data is shown to support this statement.

This information is based on three references from Matthew STATE’s group at UCSF (Sanders et al., 2015; Willsey et al., 2018; Satterstrom et al., 2020).

We indicate (page 17):

. These five genes are considered to confer an ~20-fold increase in risk, within a group of 26 genes (Sanders et al., 2015; Willsey et al., 2018; Satterstrom et al., 2020). In ASD, studies leveraging the statistical power afforded by rare *de novo* putatively damaging variants have identified more than 65 strongly associated genes (Sanders et al., 2015). The most deleterious variants (likely gene disrupting or LGD variants) in the highest-confidence subset of these genes ($N = 26$), as a group, increase the risk by about 20-fold, and LGD variants in the highest-confidence genes within this subset carry even greater risks (Willsey et al., 2018; Satterstrom et al., 2020).

- The sentence "The DYRK1A-DSCAM interaction is conserved from drosophila to humans." would imply a broad analysis of the interaction between the two proteins in many different organisms. Only proteins from *Drosophila* and humans are analyzed. In fact, the results for the interaction between the *Drosophila* proteins are not shown at all.

We indicate (page 18):

We then analyzed the interaction of the *Drosophila* homolog, minibrain (MNB), with DSCAM4. DSCAM4 (UniProtKB 2022_01) (1874 AA) has a transmembrane segment from AA 1626 to AA 1647, with an intracellular domain from AA 1648 to AA 1874. The SID extends from AA 1697 to AA 1841. The location of the DYRK1A-binding domain on DSCAM suggests a phylogenetically conservation between *Drosophila* and humans (**Figure 8 B**).

Materials and Methods. The authors should carefully check the section. Some issues are listed below, and I have also included some others in other parts of the Review.

We modified Materials and Methods accordingly.

- The subsection "Primary cell cultures and transfection" should be merged with the subsection "Primary cell cultures".

- In the case of HEK293 cells, 5 cells per transfection are very low numbers for the analysis performed, which is not clear when it has been used along the manuscript.

- The two subsections named "Statistical analysis" should be merged, and the authors check which one provides the right information. A test to check the normality of the data should be applied before using a parametric test or a non-parametric test.
- Some abbreviations are not properly annotated: ACSF, DCGIV, D-APV.
- What is a "cutting solution"?

These points have been corrected.

- The references cited in the Materials & Methods section are listed outside the Reference sections, is this OK? Anyway, some of them are missing: Cholay et al. 2010.

The references of the Materials & Methods have been added to main list of References.

Introduction.

- A reference for the incidence of DS should be provided: world-wide?, developed countries?
- DS-associated trisomy 21 could affect the whole chromosome or be partial. Please amend accordingly.

These two points have been modified accordingly, in the introduction.

- Which is the source of the 234 genes present in human chromosome 21?. ENSEMBL lists currently a total of 660 genes: 236 coding genes and 424 non coding genes. Why the authors are only referring to the coding genes as responsible for the DS-associated phenotypes? There are no reasons for such assumption.

We now indicate (page 4):

This chromosome carries 235 protein-coding genes and 441 non-protein-coding genes (Ensembl release 106 – April 2022).

- Not clear to me which are the reasons underlying the choice of only 3 references

(Ahn, Thomazeau, Smith) among the many available to sustain the involvement of the DYRK1A gene as a contributor to some of the neuropathological traits of DS.

We added the review reference from Heralut's group: Atas-Ozcan et al., 2021.

Results, Subsection 1: As mentioned above, the data of this subsection is unnecessary in the context of the rest of the manuscript and it does not reach the level of quality of the other parts of the manuscript.

- The title mention "quantitative proteomics" but no data is included.

The mention "quantitative proteomics" was removed.

- It will be very informative to provide an updated list of the mouse genes included in the Dp(16)1Yey transgenic mice and their corresponding human orthologs, so the readers have clear information on which are the 119 genes triplicated. This will help the reader to navigate among the different gene contexts of the many DS mouse models. The reference for this transgenic model is Li et al, 2017, PMID:17412756, and not Yu et al., 2010.

A recent paper of Aziz in its Suppl. Figure 1 illustrates the Dp(16)1Yey triplication.

We indicates (page 4):

The second model was a transgenic mouse line (Dp(16)1Yey) carrying a triplication of ~23.3 Mb from *Mus musculus* chr16 (Mmu16) syntenic to 115 coding-genes from HSA21 (Li et al., 2007; Aziz et al., 2018), including *DYRK1A*, precisely reflecting the gene dosage of HSA21 orthologs.

We indicates this Supplementary Figure in our text (page 6). The reference Li et al., 2017 was added instead of Yu et al., 2010.

- Information on the preparation of the libraries for RNA-Seq as well as for the type of sequencing performed (single-end?, paired-end?, length of reads?) should be provided. Which mice are used as controls in the RNA-Seq experiments? euploid littermates? Do the pools contained male and female animals? or are gender specific?

A detailed paragraph for RNA-seq method was added in Material & Methods (page 40).

We used embryonic hippocampi from transgenic mice and their littermates. They are pools contained male and female animals.

- Information on the statistical analysis used for the differential analysis of the RNA-Seq data and the parameters applied for the cut-off should be provided.

Information is now present in the Material & Methods paragraph.

We used a FDR <0.05 (see page 6: with a false discovery rate (FDR) <0.05).

- There is no proper comparison of the two set of DEG genes. They are supplied in the form of Supplementary data as pdf not easy to digest by the readers. Supp Table S1 and S2 are misassigned in the text to each of the mouse models

This was corrected (Supplementary Table S1 and Supplementary Table S2).

- The RNA-Seq experiments data are not deposited in publicly available sites.

RNA-Seq data are deposited (see page 51:

DEG data from embryonic hippocampus of 189N3 and Dp(16)1Yey/+ are available at GEO under the accession number GSE201290).

- The RNA-Seq does not detect changes in the mouse chromosome 16 triplicated genes in the Dp(16)1Yey model. A published analysis of E16 Dp(16)1Yey forebrains showed that most of the few deregulated genes with increased expression mapped to the triplicated region (Guedj et al., 2018, PMID: 27586445; Aziz et al., 2018, PMID: 29716957).

Chromosome 16 genes present in the triplicated region (Supplementary Figure S1 in Aziz et al., 2018) that are upregulated are indicated (yellow highlight) in Supplementary Table S2.

In the text, we indicate (page 6):

Furthermore, 10 genes (out of 65 upregulated genes) located in the mouse chromosome 16 syntenic region (see Figure S1 in Aziz et al., 2018) are overexpressed in our Dp(16)1Yey/+ samples, including Robo2 (**Supplementary Table S2**).

- The reference Loe-Mie et al., 2010, does not apply to the statement that "Deregulation of chromatin proteins is in agreement with published data from the authors linking changes in DYRK1A expression and alteration of chromatin-regulatory proteins.

We indicate (page 6):

This deregulation of genes encoding chromatin-related proteins is consistent with our previously reported data (Lepagnol-Bestel et al., 2009)

- The sentence "Deregulation of chromatin proteins..." should say "Deregulation of the expression of genes encoding for chromatin proteins...". This also applies to the next sentence for Dp(16)1Yey/+.

We indicate (page 6):

Amigo2 identified gene ontology (GO) categories for the downregulated DEGs, with a deregulation of the expression of genes encoding chromatin-related proteins in 189N3 mice

- References Kleschevnikov et al., 2012; Raveau et al., 2018 refer to mouse models of DS not to DS individuals.

We now indicate:

This result is entirely consistent with the impaired excitation-inhibition balance (E-I balance) of synaptic activity in DS mouse models (Kleschevnikov et al., 2012; Raveau et al., 2018).

- The quality of Figure 1 is low, there is no explanation for the different colors, if any, and the lettering is too small to properly convey any information. I do not agree with the authors in suggesting a network of protein-protein interaction changes based on deregulation of mRNA. Changes in transcripts do not necessarily translate into changes in protein amounts. Second, more than a connection among the proteins

encoded by the mRNAs, the DEGs should illustrate altered signaling pathways with specific transcription factors that use these genes as transcriptional targets. An example of such regulation can be found in the analysis of the adult hippocampus of the DS Tc1 mouse model (Granno et al., 2019; PMID: 31086297). In my opinion, the authors have chosen not the best representation of their findings for differential GO term enrichments in the two mouse models analyzed. The Programme and the parameters used for the GO analysis should be indicated in the Materials and Methods section.

We now indicate (page 44):

To complement the DAPPLE analysis, we used the WebGestalt suite (Liao et al (Liao et al., 2019), [String: functional protein association networks \(string-db.org\)](https://string-db.org) and Syngo: Synaptic Gene Ontologies and annotations consortium - An evidence-based, expert-curated resource for synapse function and gene enrichment studies (Koopmans et al., 2019).

- Figure 2 is never mentioned in the text.

We now indicate (page 7):

Using the curated ontology of the SynGO - Synaptic Gene Ontologies and annotations (Koopmans et al., 2019), we further examined the synaptic signal and found . We analyzed the 77 downregulated DEGs. 25 out of 77 DEGs were mapped to 25 unique SynGO annotated genes. The enriched cellular component ontology terms are: Synapse (n=23) $p=9.42 \times 10^{-11}$, Presynapse (n=14) $p=6.22 \times 10^{-8}$ and Postsynapse (n=12) $p=1.38 \times 10^{-5}$ (**Figure 2**).

- The authors mention in the Discussion the uniqueness of the approach in comparing differential gene expression in DS mouse models. However, at least two

published articles have already done that (Guedj et al., 2018, PMID: 27586445; Aziz et al., 2018, PMID: 29716957).

We do not mention the uniqueness of the approach in the Discussion.

Results, Subsection 2:

- A list of the baits for the primary (chromosome 21 baits) and secondary (rebounds baits) Y2H screens should be included with an entry number of the cDNA (NCBI or Ensembl ID) used for each of the genes to provide information on the specific isoform cloned for each gene. Likewise, the reader will greatly benefit for having the results of the Y2H interactions in a workable format (for instance, Excel with paired interactions). The information of the two tables will be a valuable source for many researchers.

We added a Supplementary Table that list all the interactions analyzed, including their scores and the binding site (in AA) on prey (Supplementary Table S3).

- A clear description of the categories a-f shown in Figure 3 should be provided in the Figure Legend. There are mistakes in the assignment of the letters of the individual panels, please check. The authors should make clear in the text that most of the novel interactions detected belong to the "lowest confidence" category, while the trend reversed when the already described baits are analyzed. Indeed, there is not experimental validation by orthogonal approaches providing some numbers for the expected percentage of non-specific interactions in this category. Therefore, a note of caution should be included.

The interactions with their scores are indicated in the Supplementary Table S3. Interactions with scores a to d have been validated (i.e. HUNK-AGAP3 (a) and HUNK-LIMK1 (d)).

We added in the text (pages 8-9):

These interactions were ranked by category (a to f), with a Predicted Biological Score (PBS). PBS is computed as an e-value and thresholds are attributed to define categories from high confidence (A) to lower confidence (D) interactions. The PBS e-value ranges from 0 to 1 and has been classified in five distinct categories: a to e. Inter-category thresholds were chosen

manually with respect to a training data set containing known true-positive and false-positive interactions: $a < 1e-10 < b < 1e-5 < c < 1e-2.5 < d < 1$. Complete statistical analysis of the interactome leads to the identification of highly connected interacting domain for which the corresponding PBS has been set to 1. PBS f also set to one are experimentally validated false positive (interaction with the DNA binding domain) (Formstecher et al., 2005).

- The Figure legend has to include definitions for S2, S10 and S11 used in panels E and F. A definition in the text will be also welcomed. In fact, Supplementary Table S3, including the lists of the genes, has a format non-informative at all. What does X21 refer to? Which is the statistical test used for calculating the p-value for the enrichment in Figure 3E and F?

S2, S10 and S11 are the ensembles of proteins linked to Intellectual Disabilities from list S2, S10 and S11, respectively (Gilissen et al., 2014; Deciphering Developmental Disorders Study, 2015). These lists were crossed with our list of direct interactors of baits or direct interactors of baits plus rebounds (Figure 3) (Supplementary Table S4).

X21 was corrected by HSA21

- As a piece of valuable information, the authors should include the list of genes that are responsible for the enrichment in categories shown in Figure 4, as a supplementary Excel Table. The Programme used for the analysis is missing in the Materials and Methods section.

The cross-lists are indicated in our new Supplementary Table S4.

In the M & Methods: data analysis section, we added (page 44)

The analysis of contingency tables was done using a Fisher's exact test.

- The authors might mean "We performed" instead of "We realized" (page 11, last paragraph).

This was corrected.

- Does Figure 4 include only the chromosome 21 baits? If so, it should be indicated.

We now indicate (page 9):

We performed a biological process analysis with GO DAVID (see methods) on direct interactors of both HDA21 baits and their rebounds (Figure 4).

- Within the Y2H experiments subsection in the Materials and Methods section a determination of Orthologs collection is mentioned. With what intention? how were they determined?.

For each gene, orthologs across mammals were determined. This sentence was removed

Results, Subsection 3: Although the information included in this subsection represents the validation of some of the detected YTH interactions by an independent approach, it acts a little bit as a disruptor for the main story line, which focuses on events occurring at the synapse. The two examples included in the subsection relate to putative nuclear interactions.

Nuclear interactors: this part was rewritten (pages 11-12)

- The authors could add the better known name of DROSHA for RNASEN (this would make understandable the sentence "The complex is minimally composed of the ribonuclease enzyme Drosha"), and Clusterin for CLU.

DROSHA/RNASEN is now indicated in the text (page 12) and in the legend of Figure 5 (pages 31-32).

- In Figure 5A, the authors could include arrows to indicate the direction of the interactions detected in the Y2H screen (double arrow in case, the interaction was confirmed in the rebound screens).

Arrows are now added to indicate the direction of the interactions (Figure 7, pages 35-36).

Please note that TO-PRO-3 stains DNA, not nuclear bodies. The big nuclear stain dots correspond to heterochromatin, highly evident in mouse cells.

We indicate:

In the legend of Figure 5 (page 32):

heterochromatin was labelled using Topro3 (blue fluorescence).

In Material & Methods;

Nuclear PLA interaction number was manually counted inside the heterochromatin and normalized with the nuclear area of each neuron (page 42)

- Page 16, last paragraph, it should say For DYRK1A-RNASEN interaction.
- I do not agree with the hypothesis raised by the authors in which DGRC8 haploinsufficiency may occur by titration of its partner RNASEN: first, the co-IP shown by the authors could be interpreted either by the formation of trimolecular complexes, or by the formation of independent paired interactions (RNASEN-DGRC8, RNASEN-DYRK1A) and at this point no competition among the three proteins has been demonstrated. In fact, the relative enrichment of the proteins in the IP experiment indicates that DYRK1A participate in a small pool of the RNASEN interactions in comparison with those established with DGRC8. I would eliminate the sentence, and leave the one regarding to putative links to alterations in the microRNA network, which might be a functional consequence of any type of alterations in the biology of RNASEN as result of DYRK1A physical interaction (interaction with partners, activity, stability).

We now indicate (page 12):

DYRK1A overexpression would be expected to affect the function of the *DYRK1A-RNASEN-DGRC8* interactome.

- Figure 5C and D are also included as Supplementary Figure S1A, B and C. Likewise, Figure 5A and B are also included as Supplementary Figure S3.

These points were corrected.

Results, Subsection 4: This subsection offers a nice piece of information providing validation of the interactions of different chromosome 21 encoded proteins with important players at the synapsis, highlighting two important aspects: the validation of some of the direct interactions detected by the Y2H in a physiological context and with endogenous proteins and the highlight of the synapse as an important physiological target of chromosome 21 encoded proteins.

- As in the case of Figure 5, the panel A in Figure 6 could include arrows to indicate the direction of the detected interactions (double for those cases shown in both directions). The label of y-axis in the two bar graphs included in Figure 6 is missing.

The panel A of the Figure 6 is now the Figure 7 (pages 35-36). Arrows are added to indicate the direction of the detected interactions. DYRK1A-FAM53C double interaction is also illustrated by a double arrow (Supplementary Figure S4)

Label of y-axis in Figure 6 are now included (Pages 33-34)

- Not all the PLA data shown in Figure 6 has been quantified and they should. In fact, some of the images do not allow to distinguish whether some interactions are positive or not (e.g., KNCJ6-DLG4, HUNK-SYNPO or ITSN1-DLGAP1).

We added two Supplementary Figures: the supplementary Figure S8 for control PLAs and the supplementary Figure S9 for quantifications of PLAs.

- The authors mention Supplementary Figure S1 in regard to data shown in Figure 6D, which includes no such data. Likewise, Supplementary Figure S2 does not contain data related to Figure 6G-H.

This point was corrected.

- Which is the explanation for the increase PLA signal shown in Figure 6D for KNCJ6-

DLG2 interactions in the DS mouse model: increased KCNJ6 protein expression because of the triplication of the gene? increased amounts of both partners?

KCNJ6 is triplicated but not found upregulated in our analysis (Supplementary Table S2). However, it is possible that a small upregulation is sufficient to generate an increase in the number of KCNJ6 proteins at the synapse, leading to an increase in KCNJ6-DLG2 interactions, if the concentration of DLG2 is not a limiting factor.

- Have any of the YTH interactions analyzed by PLA not been validated?

Quantifications of PLA are now illustrated in Supplementary Figure S9.

In page 13, we indicated:

We were able to detect 21 PPIs in the dendritic spine, 20 of which were novel (GRIK1-HCN1; GRIK1-KCNQ2; GRIK1-SEPT7; GRIK1-KALRN; GRIK1-DLG4; HUNK-AGAP3; HUNK-SYNPO; HUNK-LIMK1; TIAM1-BIN1; TIAM1-DLG1; KCNJ6-DLG1; KCNJ6-DLG4; KCNJ6-DLG2; ITSN1-SNAP25; ITSN1-DLGAP1; DSCAM-DLG4; DSCAM-DLG2; SIPA1L1-DLG4; DLG2-GRIN2A; DLG2-GRIN2B), the only interaction having already been documented in BioGrid but not validated at the synapse is SIPA1L1-DYRK1A (**Supplementary Figure S9**).

Results, Subsection 5:

- How the authors know that the interacting region of DSCAM and DSCAML1 with DYRK1A maps to the intracellular domain? (the one shown in Figure 7C, I assume).

The experimental data have to be shown.

- How the authors know that the corresponding orthologs in Drosophila do interact?

The experimental data has to be shown.

We detailed our experimental strategy in page 18:

The Y2H screens used here made it possible to identify the domain of the prey interacting with the bait. Once positive clones had been identified, overlapping prey fragments derived from the same gene were clustered into families. The sequence common to these fragments defines the selected interacting domain (SID) (Formstecher et al., 2005).

DYRK1A interaction occurs in the same ~90-amino acid (AA) domain of the cell adhesion molecules encoded by *DSCAM* and its paralog *DSCAML1* (**Figure 8A**). *DSCAM* (UniProtKB-O60469) is a 2012 AA protein with an extracellular domain (positions 18 to 1595), a transmembrane domain (1596-1616) and a cytoplasmic domain (1617-2012). The *DSCAM* domain interacting with DYRK1A was identified as lying between positions 1761 and 1850 AA (90 AA). *DSCAML1* (UniProtKB - Q8TD84) is a 2053 AA protein that also has an extracellular domain (positions 19 to 1591), a transmembrane domain (1592-1612) and a cytoplasmic domain (1613-2053). The domain of *DSCAM* interacting with DYRK1A was identified as lying between positions 1823 and 1907 (84 AA). We then analyzed the interaction of the *Drosophila* homolog, minibrain (MNB), with *DSCAM4*. *DSCAM4* (UniProtKB 2022_01) (1874 AA) has a transmembrane segment from AA 1626 to AA 1647, with an intracellular domain from AA 1648 to AA 1874. The SID extends from AA 1697 to AA 1841. The location of the DYRK1A-binding domain on *DSCAM* suggests a phylogenetically conservation between *Drosophila* and humans (**Figure 8 B**).

- The panels in Figure 7 are wrongly mentioned along the corresponding text in the manuscript. It looks like that the text refers to a different layout of the Figure.

This point was corrected.

- Is the *DSCAM*-DYRK1A interaction in the synapsomal fraction detected in a reverse co-IP (IP with an anti-*DSCAM* antibody)? Thus, the authors might prove the presence of *DSCAM* and DLG complexes using the co-IP method and the synaptosomes as fraction. This will provide some qualitative information on the relative presence of these proteins in association with *DSCAM*.

We obtained a limited amount of a non-commercial anti-DSCAM antibody (from Trosha Dwyer; University of Toledo) (Supplementary Table 6). This anti-DSCAM antibody has been fully validated (Purohit AA, Li W, Qu C, Dwyer T, Shao Q, Guan KL, Liu G. Down syndrome cell adhesion molecule (DSCAM) associates with uncoordinated-5C (UNC5C) in netrin-1-mediated growth cone collapse. *J Biol Chem.* 2012 Aug 3; 287(32):27126-38.). Because of the limited amount of antibody, we were not able to plan reverse co-IP experiments.

- The methods for the preparation of synaptosomes should be included in the Materials and Methods section.

We added a paragraph in Materials and Methods section (page 43)

Preparation of synaptosomes and protein extraction

Cortex from 3-4 months mice brains were dissected and homogenized (pool of six animals) in H buffer (0.32M sucrose, 5mM Hepes 1M pH7.4, 1mM EDTA) using a glass potter. The homogenate was centrifuged at 800g for 7mn to remove nuclei and debris, the supernatant was centrifuged at 9,200g for 10mn to remove cytosolic supernatant. The pellet was resuspended in H buffer and gently stratified on a discontinuous Percoll gradient (5%, 10% and 23% v/v in H-buffered percoll) and centrifuged at 20,000g for 11mn. The layer between 10% and 23% Percoll (synaptosomal fraction) was collected and washed in H buffer by centrifugation. The synaptosomal pellets were resuspended in MLB buffer (1%NP40, 100mM NaCl, 20mM Tris pH7.4 in PBS with 1x protease and phosphatase inhibitor cocktail) for 10 min on ice and centrifuged 15 min at 10,000g at 4°C. The supernatants were stored at -80°C until used and lysate protein concentration was determined using the DCTM Protein assay (Biorad).

- The authors stress the fact that DSCAM and DYRK1A interact in the postsynapse (Results and Discussion). However, I understand that the subcellular fraction used for the co-IP experiments corresponds to whole synaptosomes thereby including pre- and post-synapses. If the fractionation method renders mostly postsynapses, the enrichment should be experimentally proven. If not, the text needs to be modified.

We fully agree that synaptosomes include both pre and postsynapse. Our results indicate that DSCAM-DYRK1A can be either presynaptic or postsynaptic.

We indicated in the text (page 19):

We found that DSCAM interacted with DLG1, DLG2 and DLG4, which are *bona fide* dendritic spine components, in both Y2H approaches and PLA (**Figure 6G-H**). We also found that DYRK1A interacted with STX1A, a *bona fide* presynaptic protein (**Supplementary Figure S3**).

Overall, our results suggest that DSCAM-DYRK1A interactions may occur in both presynaptic and postsynaptic positions.

In the discussion (page 20) we indicate:

In particular, we were able to demonstrate that both DYRK1A and DSCAM can be present at the synapse.

Results, Subsection 6: Which are the differences between Figure 8 and Figure 6A? Figure 8 looks like just a repetition of that panel. In fact, this subsection is a kind of summary of results that does not stand as a Results paragraph in itself. It could be included as part of the subsection 4 or the Discussion.

We removed part A of the Figure 6.

The 33-protein network is now illustrated in the new Figure 7 (pages 35-36).

Reviewer #2 (Comments to the Authors (Required)):

This ms by Viard et al. aims to decipher pathways associated to cognitive impairment phenotypes in two different DS mice models: Dp(16)1Yey, with an extra copy of the

entire Hsa21 syntenic region, and 189N3 with an extra copy of Dyrk1A gene. Transcriptomics analysis seems to indicate different activated pathways for each genotype: chromatin related genes for the Dyrk1A model and a synapse related genes for the Dp(16)1Yey model. Y2H and further PLA analysis identified Chr21 proteins in PSD that correlate with intellectual disabilities in DS. Since the main focus of the paper is to decipher intellectual disabilities related pathways in DS models there is a considerable amount of data that need to be included

These are the main concerns:

Figure 1

The described interactomes were constructed including all the deregulated genes. Perhaps a separated analysis of unregulated and down regulated genes could give a more informative perspective

Down-regulated DEGs are now presented in new Figure 1 and Figure 2.

Figure 2

Several Dp16 up regulated genes were associated to synaptic transmission and glutamate receptor pathway. There are also 65 down regulated genes: were they related to any particular pathway?

Down-regulated DEGs are now presented in new Figure 1 and Figure 2.

Figure 3 and 4

Due to the main focus in DS models, figures 3 and 4 should be included as supplementary data, despite the huge amount of data and effort.

We want to maintain Figure 3 and 4 as informative of the Y2H effort

Figure 5

In PLA strategy, DSCR9 was expressed as a GFP fusion protein since there are no antibodies available. That is probably due to the fact that DSCR9 is described as a regulatory lncRNA.

DSCR9 is considered as coding in Takamatsu K, Maekawa K, Togashi T, Choi DK, Suzuki Y, Taylor TD, Toyoda A, Sugano S, Fujiyama A, Hattori M, Sakaki Y, Takeda T. Identification of two novel primate-specific genes in DSCR. DNA Res. 2002 Jun 30;9(3):89-97.

DSCR9 is expressed preferentially in testis and encode functionally unknown proteins of 149 amino acid residues (see Figure 1B in Takamatsu et al., 2002).

Figure 6

It is not explained what mice genotype is the source of the samples used to generate the data B to G. Then, there is a change in the nomenclature in figure legend with no link to the figure. The general organization of the data is confusing.

Described PPI need to be quantified in both DS models in order to test their relevance in ID.

Figure 7

Similar to the previous figure, DSCAM-DYRK1A interaction need to be addressed in both DS models.

Figure 8

Should either be included as a supplementary figure and commented directly in the discussion or stay at the Results section and be confronted with quantifications in both DS models.

We consider that this manuscript is a resource manuscript that cannot address all questions linked to HSA21 mouse models such as 189N3 and Dp(16)1Yey models. Results that we presented for Dp(16)1Yey (Figure 6 C-D) are only indicative that it is feasible to apply our approach to mouse model primary neuronal cultures. Analysis and quantification of PPIs using PLAs are indeed an important question but are out of scope of this resource manuscript.

Reviewer #3 (Comments to the Authors (Required)):

This study by Viard et al., studied two mouse models of Down syndrome using exome sequencing, large-scale yeast-two-hybrid and proximity ligation assays. The data point to the involvement of chromatin remodeling proteins and synaptic proteins in Down syndrome. These datasets can be of relevance for future studies aimed to better understand Down syndrome. Overall, the datasets seem robust and the comparison between the two mouse models is interesting, but the provided data is merely descriptive. The manuscript is quite dense and at times difficult to read. It could benefit from editing to make the flow more coherent. For instance, the results section contains considerable discussion and speculation that could be omitted and left for the discussion.

I have a number of concerns with the PLA assays that should be validated more extensively to be convincing:

- the phalloidin staining in these experiments gives little context, individual spines are difficult to distinguish. This makes it hard to conclude that these interactions are indeed taking place in spines. PLA spots seem to localize throughout the dendrites, many PLA spots even seem to localize to extracellular sites (e.g., KCNJ6-DLG2 in Figure 6D).
- the quality of the PLA experiment relies heavily on the quality of the antibodies that are used. To what extent are these antibodies validated?

The Supplementary Table S6 indicates the antibodies used in the study. All of them have been validated by peer-reviewed manuscripts.

- what are the negative controls in the different panels? This could be indicated more clearly.

Negative controls are shown in the new supplementary Figure S8. PLA quantifications are now illustrated in supplementary Figure S9.

- related to all these points, quantification of PLA spots seems critical to convey the specificity of this analysis. The bar graph in 6G might do this for a few, but since it

has no label it is difficult to interpret.

This point is now corrected.

Page 16: "EK293" > "HEK293" This was corrected.

We completely reorganized Supplementary Figures

Supplementary Figure S1. Protein-Protein Interaction Network generated from proteins encoded by deregulated genes identified in E17 hippocampus of 189N3 and Dp(16)1Yey transgenic mouse models, respectively.

Supplementary Figure S2. Products of deregulated genes in Dp(16)1Yey/+ are enriched in proteins linked to glutamate receptor signaling pathway and in proteins involved in an ARC-PSD95 complex linked to ID and intelligence.

Supplementary Figure S3. Analysis of the STX1A-DYRK1A and LIMK1-HUNK interactions

Supplementary Figure S4. Genomic locus including FAM53C, KDM3B and Snp 35131895.

Supplementary Figure S5 Nuclear protein-protein interactions: DYRK1A-E300, DYRK1A-CREBBP and DYRK1A-FAM53 interaction.

Supplementary Figure S6. KEGG pathway (hsa04720; Long-term potentiation - Homo sapiens) involving EP300 and CREBBP.

Supplementary Figure S7. Nuclear protein-protein interaction: unedited and high resolution data from gels used in Figure 5C.

Supplementary Figure S8. Protein-protein interactions: Control interactions

Supplementary Figure S9. Quantification of protein-protein interactions in dendritic spines

The first two Supplementary Figures are the Figures 1 and 2 of the previous version.

We added information about interaction between STX1A and LIMK1, two genes located in the Williams-Beuren deletion (Supplementary Figure S3).

KEGG pathway on synaptic LTP that involves CREBBP and EP300 was added as Supplementary Figure S5. Genomic locus including FAM53C and associated to Autism Spectrum Disorders was added as Supplementary Figure S6.

Controls PLAs and PLA quantifications were added as Supplementary Figure S8 and S9, respectively.

June 7, 2022

RE: Life Science Alliance Manuscript #LSA-2021-01205-TR

Prof. Michel simonneau
ENS Paris-Saclay
LuMin
Avenue des Sciences
Gif sur Yvette 91190
France

Dear Dr. Simonneau,

Thank you for submitting your revised manuscript entitled "Chr21 protein-protein interactions: enrichment in products involved in ID, autism and LOAD". We would be happy to publish your paper in Life Science Alliance pending final revisions necessary to meet our formatting guidelines.

- Please address the final remaining Reviewer 1 and 3 comments
- please consult our manuscript preparation guidelines <https://www.life-science-alliance.org/manuscript-prep> and make sure your manuscript sections are in the correct order
- please upload both your main and supplementary figures as single files and add a separate figure legend section to your manuscript
- please introduce the panels in your Figure legends in alphabetical order and add a Panel E to your Figure legend for Figure S3
- please add ORCID ID for both corresponding authors-you should have received instructions on how to do so
- please add the Twitter handle of your host institute/organization as well as your own or/and one of the authors in our system
- please make sure that all author names are correctly entered in our system and that the author order in the manuscript matches the author order in our system
- please add a conflict of interest statement to the main manuscript text
- please rename the accession code section into Data Availability

Figure Check:

- please expand Figure legend for Figure S8 in the explanation of the panels
- Figure 5C, Row DYRK1A looks like a splice after the 2nd blot. Please provide source data for this figure

A. FINAL FILES:

- An editable version of the final text (.DOC or .DOCX) is needed for copyediting (no PDFs).
- High-resolution figure, supplementary figure and video files uploaded as individual files: See our detailed guidelines for preparing your production-ready images, <https://www.life-science-alliance.org/authors>
- Summary blurb (enter in submission system): A short text summarizing in a single sentence the study (max. 200 characters)

including spaces). This text is used in conjunction with the titles of papers, hence should be informative and complementary to the title. It should describe the context and significance of the findings for a general readership; it should be written in the present tense and refer to the work in the third person. Author names should not be mentioned.

B. MANUSCRIPT ORGANIZATION AND FORMATTING:

Sincerely,

Reviewer #1 (Comments to the Authors (Required)):

I congratulate the authors for their efforts in presenting this new version of the manuscript, in which all my queries and concerns have been answered and/or amended. The manuscript now conveys the message in a clear way and every statement is supported by the experimental data. In my opinion, and in contrast to the previous version, the thorough analysis of the RNA-seq data from the two mouse models included in the revised version now qualifies it to be part of the manuscript.

Some minor questions or comments:

- In the Abstract, "We showed that a DSCAM intracellular domain bound either DYRK1A or DLGs, which are multimeric scaffolds bringing together receptors, ion channels, and associated signaling proteins".
Please check the sentence. To which protein as "a multimeric scaffold" are the authors referring?

- Figure S4C, there must be a labeling mistake in the first lane of each blot, which I understand should be "Input"

- For the PLA quantification plots in Figure S3, S4, 6, S9: In the bar plots, which is the meaning of the orange, green and blue colors in the bars?? Negative interactions for green? And orange? In the case green bars represent background values, the DYRK1A-SIPA1L1 interaction is not significantly different from the control (Fig. S9) and the DYRK1A-DLG4 interaction data is not shown in PLA assays. Therefore, the sentence "We found evidence of direct interactions between SIPA1L1 and DYRK1A or DLG4 (Figure 6G), and between DYRK1A and DLG4." likely refers to the YTH data, and not to Figure 6G.

- Page 11 "We identified DYRK1A-FAM53C interaction in human brain (Supplementary Figure S4A)". Maybe the authors refer to "Supplementary Table S4".

Reviewer #3 (Comments to the Authors (Required)):

The revised manuscript is strongly improved. However, I do think the writing could still benefit from editing to correct spelling and grammar. For instance, first line of the introduction: ".the commonest.." is incorrect > "most common". The text is also still very dense in terminology, and the presentation of results is often mixed with discussion of the results which makes it at times hard to distinguish between new findings and speculation.

Most of my concerns have been addressed adequately, but my previous concern remains unaddressed:

"the phalloidin staining in these experiments gives little context, individual spines are difficult to distinguish. This makes it hard to conclude that these interactions are indeed taking place in spines. PLA spots seem to localize throughout the dendrites, many PLA spots even seem to localize to extracellular sites (e.g., KCNJ6-DLG2 in Figure 6D)."

Without higher resolution images, additional experiments with markers for the PSD, and specific analysis, the presented data do not provide any evidence that the interactions are localized/enriched in spines or at synapses. With the presented data it can only be concluded that the interactions take place in dendrites and the conclusions in the text should be adjusted accordingly:

- Page 13: "Interactome of HSA21 proteins located in the postsynaptic compartment of the dendritic spine: [...] > "Interactome of HSA21 proteins located in dendrites..."
- Page 13: "We were able to detect 21 PPIs in the dendritic spine,..." > " We were able to detect 21 PPIs in dendrites,"
- Page 14: "These three interactions were validated at the synapse by PLA" > "These three interactions were validated in dendrites by PLA"
- Page 15: "PLA provided evidence for TIAM1-BIN1 and TIAM1-DLG1 interactions in the dendritic spines" > "PLA provided evidence for TIAM1-BIN1 and TIAM1-DLG1 interactions in dendrites"

Figures S8 and S9 are important additions and strengthen the dataset. It could be indicated more clearly that these are negative controls. In Figure S9 the y-axes of the graphs should be labeled.

Below a list of suggested textual corrections. This list is however probably not complete. Thorough editing of the text to check for spelling and grammar would greatly benefit this manuscript.

- Page 5: "in the dendritic spine postsynaptic density" is never used as such, perhaps this would be better: "in the postsynaptic density in dendritic spines".
- Page 6: "By contrast," > "In contrast"
- Page 19: "immunochemical" > "immunocytochemical"
- Page 19: "This preclude their use" > "this precludes their use"

July 6, 2022

RE: Life Science Alliance Manuscript #LSA-2021-01205-TRR

Prof. Michel simonneau
ENS Paris-Saclay
LuMin
Avenue des Sciences
Gif sur Yvette 91190
France

Dear Dr. simonneau,

Thank you for submitting your Research Article entitled "Chr21 protein-protein interactions: enrichment in products involved in ID, autism and LOAD". It is a pleasure to let you know that your manuscript is now accepted for publication in Life Science Alliance. Congratulations on this interesting work.

DISTRIBUTION OF MATERIALS:

Again, congratulations on a very nice paper. I hope you found the review process to be constructive and are pleased with how the manuscript was handled editorially. We look forward to future exciting submissions from your lab.

Sincerely,
